# Non-stationary Bandits with Knapsacks

**Shang Liu**[†]   **Jiashuo Jiang**[‡]   **Xiaocheng Li**[†]
[†]Imperial College Business School, Imperial College London
[‡]NYU Stern School of Business
s.liu21@imperial.ac.uk, jj2398@stern.nyu.edu, xiaocheng.li@imperial.ac.uk

## Abstract

In this paper, we study the problem of bandits with knapsacks (BwK) in a non-stationary environment. The BwK problem generalizes the multi-arm bandit (MAB) problem to model the resource consumption associated with playing each arm. At each time, the decision maker/player chooses to play an arm, and s/he will receive a reward and consume certain amount of resource from each of the multiple resource types. The objective is to maximize the cumulative reward over a finite horizon subject to some knapsack constraints on the resources. Existing works study the BwK problem under either a stochastic or adversarial environment. Our paper considers a non-stationary environment which continuously interpolates these two extremes. We first show that the traditional notion of variation budget is insufficient to characterize the non-stationarity of the BwK problem for a sublinear regret due to the presence of the constraints, and then we propose a new notion of global non-stationarity measure. We employ both non-stationarity measures to derive upper and lower bounds for the problem. Our results are based on a primal-dual analysis of the underlying linear programs and highlight the interplay between the constraints and the non-stationarity. Finally, we also extend the non-stationarity measure to the problem of online convex optimization with constraints and obtain new regret bounds accordingly.

## 1 Introduction

The multi-armed bandit (MAB) problem characterizes a problem for which a limited amount of resource must be allocated between competing (alternative) choices in a way that maximizes the expected gain. The *bandits with knapsacks* (BwK) problem generalizes the multi-armed bandits problem to allow more general resource constraints structure on the decisions made over time, in addition to the customary limitation on the time horizon. Specifically, for the BwK problem, the decision maker/player chooses to play an arm at each time period; s/he will receive a reward and consume certain amount of resource from each of the multiple resource types. Accordingly, the objective is to maximize the cumulative reward over a finite time horizon and subject to an initial budget of multiple resource types. The BwK problem was first introduced by Badanidiyuru et al. [2013] as a general framework to model a wide range of applications, including dynamic pricing and revenue management [Besbes and Zeevi, 2012], Adwords problem [Mehta et al., 2005] and more.

The standard setting of the BwK problem is stochastic where the joint distribution of reward and resource consumption for each arm remains stationary (identical) over time. Under such setting, a linear program (LP), that takes the expected reward and resource consumption of each arm as input, both serves as the benchmark for regret analysis and drives the algorithm design [Badanidiyuru et al., 2013, Agrawal and Devanur, 2014]. Notably, a static best distribution prescribed by the LP's optimal solution is used for defining the regret benchmark. An alternative setting is the adversarial BwK problem where the reward and the consumption may no long follow a distribution and they can be chosen arbitrarily over time. Under the adversarial setting, a sublinear regret is not achievable in

the worst case; Immorlica et al. [2019] derive a $O(\log T)$ competitive ratio against the static best distribution benchmark which is aligned with the static optimal benchmark in the adversarial bandits problem [Auer et al., 1995]. Another key of the BwK problem is the number of resource types $d$. When $d = 1$, one optimal decision is to play the arm with largest (expected) reward to (expected) resource consumption ratio, where the algorithm design and analysis can be largely reduced to the MAB problem. When $d > 1$, the optimal decision in general requires to play a combination of arms (corresponding the optimal basis of the underlying LP). Rangi et al. [2018] focus on the case of $d = 1$ and propose an EXP3-based algorithm that attains a regret of $O(\sqrt{mB \log m})$ against the best fixed distribution benchmark. Their result thus bridges the gap between the stochastic BwK problem and the adversarial BwK problem for the case of $d = 1$. The difference between the cases of $d = 1$ and $d > 1$ is also exhibited in the derivation of problem-dependent regret bounds for the stochastic BwK problem [Flajolet and Jaillet, 2015, Li et al., 2021, Sankararaman and Slivkins, 2021].

In this paper, we study the non-stationary BwK problem where the reward and the resource consumption at each time are sampled from a distribution as the stochastic BwK problem but the distribution may change over time. The setting relaxes the temporally i.i.d. assumption in the stochastic setting and it can be viewed as a soft measure of adversity. We aim to relate the non-stationarity (or adversity) of the distribution change with the best-achievable algorithm performance, and thus our result bridges the two extremes of BwK problem: stochastic BwK and adversarial BwK. We consider a dynamic benchmark to define the regret; while such a benchmark is aligned with the dynamic benchmark in other non-stationary learning problem [Besbes et al., 2014, 2015, Cheung et al., 2019, Faury et al., 2021], it is stronger than the static distribution benchmark in adversarial BwK [Rangi et al., 2018, Immorlica et al., 2019]. Importantly, we use simple examples and lower bound results to show that the traditional non-stationarity measures such as change points and variation budget are not suitable for the BwK problem due to the presence of the constraints. We introduce a new non-stationarity measure called *global variation budget* and employ both of this new measure and the original variation budget to capture the underlying non-stationarity of the BwK problem. We analyze the performance of a sliding-window UCB-based BwK algorithm and derive a near-optimal regret bound. Furthermore, we show that the new non-stationarity measure can also be applied to the problem of *online convex optimization with constraints* (OCOwC) and extend the analyses therein.

## 1.1 Related literature

The study of non-stationary bandits problem begins with the change-point or piecewise-stationary setting where the distribution of the rewards remains constant over epochs and changes at unknown time instants [Garivier and Moulines, 2008, Yu and Mannor, 2009]. The prototype of non-stationary algorithms such as discounted UCB and sliding-window UCB are proposed and analyzed in [Garivier and Moulines, 2008] to robustify the standard UCB algorithm against the environment change. The prevalent variation budget measure $V = \sum_{t=1}^{T-1} \|\mathcal{P}_t - \mathcal{P}_{t+1}\|$ (where $\mathcal{P}_t$ and the norm bear different meaning under different context) is later proposed and widely studied under different contexts, such as non-stationary stochastic optimization (Besbes et al. [2015]), non-stationary MAB (Besbes et al. [2014]), non-stationary linear bandits (Cheung et al. [2019]), and non-stationary generalized linear bandits (Faury et al. [2021]) problems. In general, these works derive lower bound of $\Omega(V^{\frac{1}{3}}T^{\frac{2}{3}})$, and propose algorithms that achieve near-optimal regret of $\tilde{O}(V^{\frac{1}{3}}T^{\frac{2}{3}})$. Cheung et al. [2019] and Faury et al. [2021] require various assumptions on the decision set to attain such upper bound; under more general conditions, a regret bound of $\tilde{O}(V^{\frac{1}{5}}T^{\frac{4}{5}})$ can be obtained [Faury et al., 2021]. With the soft measure of non-stationarity, the existing results manage to obtain sublinear regret bounds in $T$ against dynamic optimal benchmarks. In contrast, a linear regret in $T$ is generally inevitable against the dynamic benchmark when the underlying environment is adversarial. We remark that while all these existing works consider the unconstrained setting, our work complements this line of literature with a proper measure of non-stationarity in the constrained setting.

Another related stream of literature is the problem of online convex optimization with constraints (OCOwC) which extends the OCO problem in a constrained setting. There are two types of constraints considered: the long-term constraint [Jenatton et al., 2016, Neely and Yu, 2017] and the cumulative constraint [Yuan and Lamperski, 2018, Yi et al., 2021]. The former defines the constraint violation by $\|(\sum_{t=1}^{T} \boldsymbol{g}_t(x_t))^+\|$ whilst the latter defines it by $\sum_{t=1}^{T} \|(\boldsymbol{g}_t(x_t))^+\|$ where $(\cdot)^+$ is the positive-part function. The existing works mainly study the setting where $\boldsymbol{g}_t = \boldsymbol{g}$ for all $t$ and $\boldsymbol{g}$ is known a priori. Neely and Yu [2017] considers a setting where $\boldsymbol{g}_t$ is i.i.d. generated from some distribution. In

this paper, we show that our non-stationarity measure naturally extends to this problem and derives bounds for OCOwC when $\boldsymbol{g}_t$'s are generated in a non-stationary manner.

## 2  Problem Setup

We first introduce the formulation of the BwK problem. The decision-maker/learner is given a fixed finite set of arms $\mathcal{A}$ (with $|\mathcal{A}| = m$) called as *action set*. There are $d$ knapsack constraints with a known initial budget of $B_j$ for $j \in [d]$. Without loss of generality, we assume $B_j = B$ for all $j$. There is a finite time horizon $T$, which is also known in advance. At each time $t = 1, ..., T$, the learner must choose either to play an arm $i_t$ or to do nothing but wait. If the learner plays the arm $i$ at time $t$, s/he will receive a reward $r_{t,i} \in [0, 1]$ and consume $c_{t,j,i} \in [0, 1]$ amount of each resource $j$ from the initial budget $B$. As the convention, we introduce a *null arm* to model "doing nothing" which generates a reward of zero and consumes no resource at all. We assume $(\boldsymbol{r}_t, \boldsymbol{c}_t)$ is sampled from some distribution $\mathcal{P}_t$ independently over time where $\boldsymbol{r}_t = \{r_{t,i}\}_{i \in [m]}$ and $\boldsymbol{c}_t = \{c_{t,j,i}\}_{i \in [m], j \in [d]}$. In the stochastic BwK problem, the distribution $\mathcal{P}_t$ remains unchanged over time, while in the adversarial BwK problem, $\mathcal{P}_t$ is chosen adversarially. In our paper, we allow $\mathcal{P}_t$ to be chosen adversarially, while we use some non-stationarity measure to control the extent of adversity in choosing $\mathcal{P}_t$'s.

At each time $t$, the learner needs to pick $i_t$ using the past observations until time $t - 1$ but without observing the outcomes of time step $t$. The resource constraints are assumed to be *hard* constraints, i.e., the learner must stop at the earliest time $\tau$ when at least one constraint is violated, i.e. $\sum_{t=1}^{\tau} c_{t,j,i_t} > B$, or the time horizon $T$ is exceeded. The objective is to maximize the expected cumulative reward until time $\tau$, i.e. $\mathbb{E}[\sum_{t=1}^{\tau-1} r_{t,i_t}]$. To measure the performance of a learner, we define the regret of the algorithm/policy $\pi$ adopted by the learner as

$$\text{Reg}(\pi, T) \coloneqq \text{OPT}(T) - \mathbb{E}\left[\sum_{t=1}^{\tau-1} r_{t,i_t} \middle| \pi\right].$$

Here $\text{OPT}(T)$ denotes the expected cumulative reward of the optimal dynamic policy given all the knowledge of $\mathcal{P}_t$'s in advance. Its definition is based on the dynamic optimal benchmark which allows the arm play decisions/distributions to change over time.

### 2.1  A Motivating Example

The conventional variation budget is defined by

$$V \coloneqq \sum_{t=1}^{T-1} \text{dist}(\mathcal{P}_t, \mathcal{P}_{t+1}).$$

By twisting the definition of the metric $\text{dist}(\cdot, \cdot)$, it captures many of the existing non-stationary measures for unconstrained learning problems. Now we use a simple example to illustrate why $V$ no longer fits for the constrained setting. Similar examples have been used to motivate algorithm design and lower bound analysis in [Golrezaei et al., 2014, Cheung et al., 2019, Jiang et al., 2020], but have not been yet be exploited in a partial-information setting such as bandits problems.

Consider a BwK problem instance that has two arms (one actual arm and one null arm), and a single resource constraint with initial capacity of $\frac{T}{2}$. Without loss of generality, we assume $T$ is even. The null arm has zero reward and zero resource consumption throughout the horizon, and the actual arm always consumes 1 unit of resource (deterministically) for each play and outputs 1 unit of reward (deterministically) for the first half of the horizon, i.e., when $t = 1, ..., \frac{T}{2}$. For the second half of the horizon $t = \frac{T}{2} + 1, ..., T$, the reward of the actual arm will change to either $1 + \Delta$ or $1 - \Delta$, and the change happens adversarially. For this problem instance, the distribution $\mathcal{P}_t$ only changes once, i.e., $V = \Delta$ (varying up to constant due to the metric definition). But for this problem instance, a regret of $\frac{T \cdot \Delta}{4}$ is inevitable. To see this, if the player plays the actual arm no less than $\frac{T}{4}$ times, then the distributions of the second half can adversarially change to the reward $1 + \Delta$, and this will result in a $\frac{T \cdot \Delta}{4}$ regret at least. The same for the case of playing the actual arm for the case of no more than $\frac{T}{4}$ times, and we defer the formal analysis to the proof of the lower bounds in Theorem 2.

This problem instance implies that a sublinear dependency on $T$ cannot be achieved with merely the variation budget $V$ to characterize the non-stationarity. Because with the presence of the constraint(s),

the arm play decisions over time are all coupled together not only through the learning procedure, but also through the "global" resource constraint(s). For the unconstrained problems, the non-stationarity affects the effectiveness of the learning of the system; for the constrained problems, the non-stationarity further challenges the decision making process through the lens of the constraints.

## 2.2 Non-stationarity Measure and Linear Programs

We denote the expected reward vector as $\boldsymbol{\mu}_t = \{\mu_{t,i}\}_{i\in[m]}$ and the expected consumption matrix as $\boldsymbol{C}_t = \{C_{t,j,i}\}_{j\in[d],i\in[m]}$, i.e.,

$$\mu_{t,i} := \mathbb{E}[r_{t,i}], \quad C_{t,j,i} := \mathbb{E}[c_{t,j,i}].$$

We first follow the conventional variation budget and define the *local non-stationarity budget*: [1]

$$V_1 := \sum_{t=1}^{T-1} \|\boldsymbol{\mu}_t - \boldsymbol{\mu}_{t+1}\|_\infty,$$

$$V_{2,j} := \sum_{t=1}^{T-1} \|\boldsymbol{C}_{t,j} - \boldsymbol{C}_{t+1,j}\|_\infty, \quad V_2 := \max_{1\le j\le d} V_{2,j}.$$

We refer to the measure as a local one in that they capture the local change of the distributions between time $t$ and time $t+1$.

Next, we define the *global non-stationarity budget*:

$$W_1 := \sum_{t=1}^{T} \|\boldsymbol{\mu}_t - \bar{\boldsymbol{\mu}}\|_\infty,$$

$$W_2 := \sum_{t=1}^{T} \|\boldsymbol{C}_t - \bar{\boldsymbol{C}}\|_1,$$

where $\bar{\boldsymbol{\mu}} = \frac{1}{T}\sum_{t=1}^{T} \boldsymbol{\mu}_t$ and $\bar{\boldsymbol{C}} = \frac{1}{T}\sum_{t=1}^{T} \boldsymbol{C}_t$. These measures capture the total deviations for all the $\boldsymbol{\mu}_t$'s and $\boldsymbol{C}_t$ from their global averages. By definition, $W_1$ and $W_2$ upper bound $V_1$ and $V_2$ (up to a constant), so they can be viewed as a more strict measure of non-stationarity than the local budget. In the definition of $W_2$, the $L_1$ norm is not essential and it aims to sharpen the regret bounds.

All the existing analyses of the BwK problem utilize the underlying linear program (LP) and establish the LP's optimal objective value as an upper bound of the regret benchmark OPT($T$). In a non-stationary environment, the underlying LP is given by

$$\text{LP}\left(\{\boldsymbol{\mu}_t\}, \{\boldsymbol{C}_t\}, T\right) := \max_{\boldsymbol{x}_1,\ldots,\boldsymbol{x}_T} \sum_{t=1}^{T} \boldsymbol{\mu}_t^\top \boldsymbol{x}_t$$

$$\text{s.t.} \sum_{t=1}^{T} \boldsymbol{C}_t \boldsymbol{x}_t \le \boldsymbol{B}, \quad \boldsymbol{x}_t \in \Delta_m, \; t = 1,\ldots,T,$$

where $\boldsymbol{B} = (B,...,B)^\top$ and $\Delta_m$ denotes the $m$-dimensional standard simplex. We know that

$$\text{LP}(\{\boldsymbol{\mu}_t\}, \{\boldsymbol{C}_t\}, T) \ge \text{OPT(T)}.$$

In the rest of our paper, we will use $\text{LP}(\{\boldsymbol{\mu}_t\}, \{\boldsymbol{C}_t\}, T)$ for the analysis of regret upper bound. We remark that in terms of this LP upper bound, the dynamic benchmark allows the $\boldsymbol{x}_t$ to take different values, while the static benchmark will impose an additional constraint to require all the $\boldsymbol{x}_t$ be the same.

For notation simplicity, we introduce the following linear growth assumption. All the results in this paper still hold without this condition.

---

[1]Throughout the paper, for a vector $\boldsymbol{v} \in \mathbb{R}^n$, we denote its $L_1$ norm and $L_\infty$ norm by $\|\boldsymbol{v}\|_1 := \sum_{i=1}^{n} |v_i|$, $\|\boldsymbol{v}\|_\infty := \max_{1\le i\le n} |v_i|$. For a matrix $\boldsymbol{M} \in \mathbb{R}^{m\times n}$, we denote its $L_1$ norm and $L_\infty$ norm by $\|\boldsymbol{M}\|_1 := \sup_{\boldsymbol{x}\neq 0} \frac{\|\boldsymbol{M}\boldsymbol{x}\|_1}{\|\boldsymbol{x}\|_1} = \max_{1\le j\le n} \sum_{i=1}^{m} |M_{ij}|$, $\|\boldsymbol{M}\|_\infty := \sup_{\boldsymbol{x}\neq 0} \frac{\|\boldsymbol{M}\boldsymbol{x}\|_\infty}{\|\boldsymbol{x}\|_\infty} = \max_{1\le i\le m} \sum_{j=1}^{n} |M_{ij}|$.

**Assumption 1** (Linear Growth). *We have the resource budget $B = bT$ for some $b > 0$.*

Define the single-step LP by

$$\text{LP}(\boldsymbol{\mu}, \boldsymbol{C}) := \max_{\boldsymbol{x}} \ \boldsymbol{\mu}^\top \boldsymbol{x}$$
$$\text{s.t. } \boldsymbol{C}\boldsymbol{x} \leq \boldsymbol{b}, \quad \boldsymbol{x} \in \Delta_m.$$

where $\boldsymbol{b} = (b, ..., b)^\top$. The single-step LP's optimal objective value can be interpreted as the single-step optimal reward under a normalized resource budget $\boldsymbol{b}$.

Throughout the paper, we will use the dual program and the dual variables to relate the resource consumption with the reward, especially for the non-stationary environment. The dual of the benchmark $\text{LP}(\{\boldsymbol{\mu}_t\}, \{\boldsymbol{C}_t\}, T)$ is

$$\text{DLP}(\{\boldsymbol{\mu}_t\}, \{\boldsymbol{C}_t\}) := \min_{\boldsymbol{q}, \boldsymbol{\alpha}} T \cdot \boldsymbol{b}^\top \boldsymbol{q} + \sum_{t=1}^{T} \alpha_t$$
$$\text{s.t. } \boldsymbol{q} \geq 0, \quad \boldsymbol{\mu}_t - \boldsymbol{C}_t^\top \boldsymbol{q} - \alpha_t \cdot \mathbf{1}_m \leq 0, \quad t = 1, \dots, T$$

where $\mathbf{1}_m$ denotes an $m$-dimensional all-one vector. Here we denotes one optimal solution as $(\boldsymbol{q}^*, \boldsymbol{\alpha}^*)$.

The dual of the single-step $\text{LP}(\boldsymbol{\mu}_t, \boldsymbol{C}_t)$ is

$$\text{DLP}(\boldsymbol{\mu}_t, \boldsymbol{C}_t) := \min_{\boldsymbol{q}, \alpha} \boldsymbol{b}^\top \boldsymbol{q} + \alpha$$
$$\text{s.t. } \boldsymbol{q} \geq 0, \quad \boldsymbol{\mu}_t - \boldsymbol{C}_t^\top \boldsymbol{q} - \alpha \cdot \mathbf{1}_m \leq 0.$$

Here we denotes one optimal solution as $(\boldsymbol{q}_t^*, \alpha_t^*)$. The dual optimal solutions $\boldsymbol{q}^*$ and $\boldsymbol{q}_t^*$ are also known as the dual price, and they quantify the cost efficiency of each arm play.

Define
$$\bar{q} = \max \left\{ \|\boldsymbol{q}^*\|_\infty, \|\boldsymbol{q}_t^*\|_\infty, t = 1, ..., T \right\}.$$

The quantity $\bar{q}$ captures the maximum amount of achievable reward by each unit of resource consumption. We will return with more discussion on this quantity $\bar{q}$ after we present the regret bound.

**Lemma 1.** *We have the following upper bound on $\bar{q}$,*

$$\bar{q} \leq \frac{1}{b}.$$

**Proposition 1.** *We have*

$$\sum_{t=1}^{T} \text{LP}(\boldsymbol{\mu}_t, \boldsymbol{C}_t) \leq \text{LP}(\{\boldsymbol{\mu}_t\}, \{\boldsymbol{C}_t\}, T) \leq T \cdot \text{LP}(\bar{\boldsymbol{\mu}}, \bar{\boldsymbol{C}}) + W_1 + \bar{q} W_2 \leq \sum_{t=1}^{T} \text{LP}(\boldsymbol{\mu}_t, \boldsymbol{C}_t) + 2(W_1 + \bar{q} W_2).$$

Proposition 1 relates the optimal value of the benchmark $\text{LP}(\{\boldsymbol{\mu}_t\}, \{\boldsymbol{C}_t\}, T)$ with the optimal values of the single-step LPs. To interpret the bound, $\text{LP}(\{\boldsymbol{\mu}_t\}, \{\boldsymbol{C}_t\}, T)$ works as an upper bound of the $\text{OPT}(T)$ in defining the regret, and the summation of $\text{LP}(\boldsymbol{\mu}_t, \boldsymbol{C}_t)$ corresponds to the total reward obtained by evenly allocating the resource over all time periods. In a stationary environment, these two are the same as the optimal decision naturally corresponds to an even allocation of the resources. In a non-stationary environment, it can happen that the optimal allocation of the resource corresponds an uneven one for $\text{LP}(\{\boldsymbol{\mu}_t\}, \{\boldsymbol{C}_t\}, T)$. For the problem instance in Section 2.1, the optimal allocation may be either to exhaust all the resource in first half of time periods or preserve all the resource for the second half. In such case, forcing an even allocation will reduce the total reward obtained. The proposition tells that the reduction can be bounded by $2W_1 + 2\bar{q} W_2$ where the non-stationarity in resource consumption $W_2$ is weighted by the dual price upper bound $\bar{q}$.

## 3  Sliding-Window UCB for Non-stationary BwK

In this section, we adapt the standard sliding-window UCB algorithm for the BwK problem (Algorithm 1) and derive a near-optimal regret bound. The algorithm will terminate when any type of the resources

is exhausted. At each time $t$, it constructs standard sliding-window confidence bounds for the reward and the resource consumption. Specifically, we define the sliding-window estimators by

$$\hat{\mu}_{t,i}^{(w)} := \frac{\sum_{s=1\vee(t-w)}^{t-1} r_{t,j} \cdot \mathbb{1}\{i_s = i\}}{n_{t,i}^{(w)} + 1}, \quad \hat{C}_{t,j,i}^{(w)} := \frac{\sum_{s=1\vee(t-w)}^{t-1} c_{t,j,i} \cdot \mathbb{1}\{i_s = i\}}{n_{t,i}^{(w)} + 1},$$

where $n_{t,i}^{(w)} = \sum_{s=1\vee(t-w)}^{t-1} \mathbb{1}\{i_s = i\}$ denotes the number of times that the $i$-th arm has been played in the last $w$ time periods. To be optimistic on the objective value, UCBs are computed for rewards and LCBs are computed for the resource consumption, respectively. With the confidence bounds, the algorithm solves a single-step LP to prescribe a randomized rule for the arm play decision.

Our algorithm can be viewed as a combination of the standard sliding-window UCB algorithm [Garivier and Moulines, 2008, Besbes et al., 2015] with the UCB for BwK algorithm [Agrawal and Devanur, 2014]. It makes a minor change compared to [Agrawal and Devanur, 2014] which solves a single-step LP with a shrinkage factor $(1 - \epsilon)$ on the right-hand-side. The shrinkage factor therein ensures that the resources will not be exhausted until the end of the horizon, but it is not essential to solving the problem. For simplicity, we choose the more natural version of the algorithm which directly solves the single-step LP. We remark that the knowledge of the initial resource budget $\boldsymbol{B}$ and the time horizon $T$ will only be used for defining the right-hand-side of the constraints for this $\mathrm{LP}(\mathrm{UCB}_t(\boldsymbol{\mu}_t), \mathrm{LCB}_t(\boldsymbol{C}_t))$.

---

**Algorithm 1** Sliding-Window UCB Algorithm for BwK

---

**Input:** Initial resource budget $\boldsymbol{B}$, time horizon $T$, window sizes $w_1$ (for reward) and $w_2$ (for resource consumption).
**Output:** Arm play indices $\{i_t\}$'s
1: **while** $t \leq T$ **do**
2:     **if** $\sum_{s=1}^{t-1} c_{t,j} > B$ for some $j$ **then**
3:         Break
4:         %% Terminate the procedure if any resource is exhausted.
5:     **end if**
6:     Construct confidence bounds $\mathrm{UCB}_t(\boldsymbol{\mu}_t), \mathrm{LCB}_t(\boldsymbol{C}_t)$ with window size $w_1, w_2$

$$\mathrm{UCB}_{t,i}(\boldsymbol{\mu}_t) := \hat{\mu}_{t,i}^{(w_1)} + \sqrt{\frac{2}{n_{t,i}^{(w_1)} + 1} \log(12mT^3)}$$

$$\mathrm{LCB}_{t,j,i}(\boldsymbol{C}_t) := \hat{C}_{t,j,i}^{(w_2)} - \sqrt{\frac{2}{n_{t,i}^{(w_2)} + 1} \log(12mdT^3)}$$

7:     Solve the single-step problem $\mathrm{LP}(\mathrm{UCB}_t(\boldsymbol{\mu}_t), \mathrm{LCB}_t(\boldsymbol{C}_t))$
8:     Denote its optimal solution by $\boldsymbol{x}_t^* = (x_{t,1}^*, ..., x_{t,m}^*)$
9:     Pick arm $i_t$ randomly according to $\boldsymbol{x}_t^*$, i.e., $\mathbb{P}(i_t = i) = x_{t,i}^*$
10:    Observe the realized reward $r_t$ and resource consumption $c_{t,j}$ for $j \in [d]$
11: **end while**

---

Now we begin to analyze the algorithm's performance. For starters, the following lemma states a standard concentration result for the sliding-window confidence bound.

**Lemma 2.** *The following inequalities hold for all $t = 1, ..., T$ with probability at least $1 - \frac{1}{3T}$:*

$$\mathrm{UCB}_{t,i}(\boldsymbol{\mu}_t) + \sum_{s=1\vee(t-w_1)}^{t-1} \|\boldsymbol{\mu}_s - \boldsymbol{\mu}_{s+1}\|_\infty \geq \mu_{t,i}, \quad \forall i,$$

$$\mathrm{LCB}_{t,j,i}(\boldsymbol{C}_t) - \sum_{s=1\vee(t-w_2)}^{t-1} \|\boldsymbol{C}_{s,j} - \boldsymbol{C}_{s+1,j}\|_\infty \leq C_{t,j,i}, \quad \forall j, i$$

*where the UCB and LCB estimators are defined in Algorithm 1.*

With Lemma 2, we can employ a concentration argument to relate the realized reward (or resource consumption) with the reward (or resource consumption) of the LP under its optimal solution. In

Lemma 3, recall that $\tau$ is the termination time of the algorithm where some type of resources is exhausted, and $\boldsymbol{x}_s^*$ is defined in Algorithm 1 as the optimal solution of the LP solved at time $s$.

**Lemma 3.** *For Algorithm 1, the following inequalities hold for all $t \leq \min\{\tau, T\}$,*

$$\left| \sum_{s=1}^{t} (r_t - \mathrm{UCB}_s(\boldsymbol{\mu}_s)^\top \boldsymbol{x}_s^*) \right| \leq 4\sqrt{T}\log(12mT^3) + 8\sqrt{\log(12mT^3)m} \cdot \frac{T}{\sqrt{w_1}} + w_1 V_1,$$

$$\left| \sum_{s=1}^{t} (c_{s,j} - \mathrm{LCB}_t(\boldsymbol{C}_{s,j})^\top \boldsymbol{x}_s^*) \right| \leq 4\sqrt{T}\log(12mdT^3) + 8\sqrt{\log(12mdT^3)m} \cdot \frac{T}{\sqrt{w_2}} + w_2 V_2 \text{ for all } j,$$

*with probability at least $1 - \frac{1}{T}$.*

We note that the single-step LP's optimal solution is always subject to the resource constraints. So the second group of inequalities in Lemma 3 implies the following bound on the termination time $\tau$. Recall that $b$ is the resource budget per time period; for a larger $b$, the resource consumption process becomes more stable and the budget is accordingly less likely to be exhausted too early.

**Corollary 1.** *If we choose $w_2 = \min\left\{ \lceil m^{\frac{1}{3}} V_2^{-\frac{2}{3}} T^{\frac{2}{3}} \log^{\frac{1}{3}}(12mdT^3) \rceil, T \right\}$ in Algorithm 1, the following inequality holds*

$$T - \tau \leq \frac{1}{b} \cdot \left( 10m^{\frac{1}{3}} V_2^{\frac{1}{3}} T^{\frac{2}{3}} \log^{\frac{1}{3}}(12mdT^3) + 8\sqrt{mT}\sqrt{\log(12mdT^3)} + 4\sqrt{T}\log(12mdT^3) \right)$$

$$= \tilde{O}\left( \frac{1}{b}(m^{1/3} V_2^{\frac{1}{3}} T^{\frac{2}{3}} + \sqrt{mT}) \right)$$

*with probability at least $1 - \frac{1}{2T}$.*

To summarize, Lemma 3 compares the realized reward with the cumulative reward of the single-step LPs, and Corollary 1 bounds the termination time of the algorithm. Recall that Proposition 1 relates the cumulative reward of the single-step LPs with the underlying LP – the regret benchmark. Putting together these results, we can optimize $w_1$ and $w_2$ by choosing

$$w_1 = \min\left\{ \lceil m^{\frac{1}{3}} V_1^{-\frac{2}{3}} T^{\frac{2}{3}} \log^{\frac{1}{3}}(12mT^3) \rceil, T \right\}, \quad w_2 = \min\left\{ \lceil m^{\frac{1}{3}} V_2^{-\frac{2}{3}} T^{\frac{2}{3}} \log^{\frac{1}{3}}(12mdT^3) \rceil, T \right\}$$

and then obtain the final regret upper bound as follows.

**Theorem 1.** *Under Assumption 1, the regret of Algorithm 1 is upper bounded as*

$$\mathrm{Reg}(\pi_1, T) \leq \frac{1}{b}\left( 4\sqrt{T}\log(12mdT^3) + (10 + 2d)m^{\frac{1}{3}} V_2^{\frac{1}{3}} T^{\frac{2}{3}} \log^{\frac{1}{3}}(12mdT^3) + 8\sqrt{mT}\sqrt{\log(12mdT^3)} + 1 \right)$$

$$+ 4\sqrt{T}\log(12mT^3) + 12m^{\frac{1}{3}} V_1^{\frac{1}{3}} T^{\frac{2}{3}} \log^{\frac{1}{3}}(12mT^3) + 2(W_1 + \bar{q}W_2)$$

$$= \tilde{O}\left( \frac{1}{b}\sqrt{mT} + m^{\frac{1}{3}} V_1^{\frac{1}{3}} T^{\frac{2}{3}} + \frac{1}{b} \cdot m^{\frac{1}{3}} dV_2^{\frac{1}{3}} T^{\frac{2}{3}} + W_1 + \bar{q}W_2 \right)$$

*where $\pi_1$ denotes the policy specified by Algorithm 1 and $\tilde{O}(\cdot)$ hides the universal constant and the logarithmic factors.*

Theorem 1 provides a regret upper bound for Algorithm 1 that consists of several parts. The first part of the regret bound is on the order of $\frac{1}{b}\sqrt{mT}$ and it captures the regret when the underlying environment is stationary. The remaining parts of the regret bound characterize the relation between the intensity of non-stationarity and the algorithm performance. The non-stationarity from both the reward and the resource consumption will contribute to the regret bound and that from the resource consumption will be weighted by a factor of $\frac{1}{b}$ or $q$ (See Lemma 1 for the relation between these two). For the local non-stationarity $V_1$ and $V_2$, the algorithm requires a prior knowledge of them to decide the window length, aligned with the existing works on non-stationarity in unconstrained settings. For the global non-stationarity $W_1$ and $W_2$, the algorithm does not require any prior knowledge and they will contribute additively to the regret bound. Together with the lower bound results in Theorem 2, we argue that the regret bound cannot be further improved even with the knowledge of $W_1$ and $W_2$.

When the underlying environment degenerates from a non-stationary one to a stationary one, all the terms related to $V_1, V_2, W_1$ and $W_2$ will disappear and then the upper bound in Theorem 1 matches

the regret upper bound for the stochastic BwK setting. In Theorem 1, we choose to represent the upper bound in terms of $b$ and $T$ so as to reveal its dependency on $T$ and draw a better comparison with the literature on unconstrained bandits problem. We provide a second version of Theorem 1 in Appendix E that matches the existing high probability bounds using $\mathrm{OPT}(T)$ [Badanidiyuru et al., 2013, Agrawal and Devanur, 2014]. In contrast to the $\Theta(\log T)$-competitiveness result in the adversarial BwK [Immorlica et al., 2019], our result implies that with a property measure of the non-stationarity/adversity, the sliding-window design provides an effective approach to robustify the algorithm performance when the underlying environment changes from stationary to non-stationary, and the according algorithm performance will not drastically deteriorate when the intensity of the non-stationarity is small.

When the resource constraints become non-binding for the underlying LPs, the underlying environment degenerates from a constrained setting to an unconstrained setting. We separate the discussion for the two cases: (i) the benchmark LP and all the single-step LPs have only non-binding constraints; (ii) the benchmark LP have only non-binding constraints but some single-step LP have binding constraints. For case (i), the regret bound in Theorem 1 will match the non-stationary MAB bound [Besbes et al., 2014]. For case (ii), the match will not happen and this is inevitable. We elaborate the discussion in Section D.

**Theorem 2** (Regret lower bounds). *The following lower bounds hold for any policy $\pi$,*

*(i)* $\mathrm{Reg}(\pi, T) = \Omega(m^{\frac{1}{3}} V_1^{\frac{1}{3}} T^{\frac{2}{3}})$.

*(ii)* $\mathrm{Reg}(\pi, T) = \Omega(\frac{1}{b} \cdot m^{\frac{1}{3}} V_2^{\frac{1}{3}} T^{\frac{2}{3}})$.

*(iii)* $\mathrm{Reg}(\pi, T) = \Omega(W_1 + \bar{q} W_2)$.

Theorem 2 presents a few lower bounds for the problem. The first and the second lower bounds are adapted from the lower bound example in non-stationary MAB [Besbes et al., 2014] and the third lower bound is adapted from the motivating example in 2.1. There are simple examples where each one of these three lower bounds dominates over the other two. In this sense, all the non-stationarity-related terms in the upper bound of Theorem 1 are necessary including the parameters $1/b$ and $\bar{q}$. There is one gap between the lower bound and the upper bound with regard to the number of constraints $d$ in the term related to $V_2$. We leave it as future work to reduce the factor to $\log d$ with some finer analysis. Furthermore, we provide a sharper definition of the global nonstationarity measure $W_1^{\min}$ and $W_2^{\min}$ in replacement of $W_1$ and $W_2$ in Appendix C2. It makes no essential change to our analysis, and the two measures coincide with each other on the lower bound problem instance. We choose to use $W_1$ and $W_2$ for presentation simplicity, while $W_1^{\min}$ and $W_2^{\min}$ can capture the more detailed temporal structure of the nonstationarity. The discussion leaves an open question that whether the knowledge of some additional structure of the environment can further reduce the global non-stationarity.

## 4    Extension to Online Convex Optimization with Constraints

In this section, we show how our notion of non-stationarity measure can be extended to the problem of online convex optimization with constraints (OCOwC). Similar to BwK, OCOwC also models a sequential decision making problem under the presence of constraints. Specifically, at each time $t$, the player chooses an action $\boldsymbol{x}_t$ from some convex set $\mathcal{X}$. After the choice, a convex cost function $f_t : \mathcal{X} \to \mathbb{R}$ and a concave resource consumption function $\boldsymbol{g}_t = (g_{t,1}, ...., g_{t,d}) : \mathcal{X} \to \mathbb{R}^d$ are revealed. As in the standard setting of OCO, the functions $f_t$ is adversarially chosen and thus a static benchmark is consider and defined by

$$\mathrm{OPT}(T) := \min_{\boldsymbol{x} \in \mathcal{X}} \sum_{t=1}^{T} f_t(\boldsymbol{x})$$

$$\text{s.t.} \sum_{t=1}^{T} g_{t,i}(\boldsymbol{x}) \le 0, \text{ for } i \in [d].$$

Denote its optimal solution as $\boldsymbol{x}^*$ and its dual optimal solution as $\boldsymbol{q}^*$.

While the existing works consider the case when $g_t$'s are static or sample i.i.d. from some distribution $\mathcal{P}$. We consider a non-stationary setting where $g_t$ may change adversarially over time. We define a global non-stationarity measure by

$$W := \sum_{t=1}^{T} \sum_{j=1}^{d} \|g_{t,j} - \bar{g}_j\|_\infty$$

where $\bar{g}_j = \frac{1}{T} \sum_{t=1}^{T} g_{t,j}$ and $\|f\|_\infty := \sup_{\boldsymbol{x} \in \mathcal{X}} |f(\boldsymbol{x})|$.

The OCOwC problem considers the following bi-objective performance measure:

$$\text{Reg}_1(\pi, T) = \sum_{t=1}^{T} f_t(\boldsymbol{x}_t) - \sum_{t=1}^{T} f_t(\boldsymbol{x}^*)$$

$$\text{Reg}_2(\pi, T) = \sum_{i=1}^{d} \left( \sum_{t=1}^{T} g_{t,i}(\boldsymbol{x}_t) \right)^+$$

where $(\cdot)^+$ denotes the positive-part function and $\pi$ denotes the policy/algorithm.

In analogous to the single-step LPs, we consider an optimization problem with more restricted constraints as

$$\text{OPT}'(T) := \min_{\boldsymbol{x} \in \mathcal{X}} \sum_{t=1}^{T} f_t(\boldsymbol{x})$$
$$\text{s.t. } g_{t,i}(\boldsymbol{x}) \leq 0, \text{ for } t \in [T], \ i \in [d].$$

Denote its optimal solution as $\boldsymbol{x}^{*'}$, and its dual optimal solution as $\boldsymbol{q}^{*'}$. The following proposition relates the two optimal objective values.

**Assumption 2.** *We assume that Slater's condition holds for both the standard OCOwC program $OPT(T)$ and the restricted OCOwC program $OPT'(T)$. We assume that $f_t, \nabla f_t, g_{t,i}$, and $\nabla g_{t,i}$ are uniformly bounded on $\mathcal{X}$ and that $\mathcal{X}$ itself is bounded. Moreover, we assume that their dual optimal solutions are uniformly bounded by $\bar{q}$, i.e.*

$$\bar{q} = \max \left\{ \|\boldsymbol{q}^*\|_\infty, \|\boldsymbol{q}^{*'}\|_\infty \right\}.$$

The following proposition relates the two optimal objective values.

**Proposition 2.** *For OCOwC problem, under Assumption 2, we have*

$$0 \leq \text{OPT}'(T) - \text{OPT}(T) \leq \bar{q}W.$$

Utilizing the proposition, we can show that the gradient-based algorithm of [Neely and Yu, 2017] achieves the following regret for the setting of OCO with non-stationary constraints. Moreover, we further extend the results and discuss in Appendix F on an oblivious adversarial setting where $g_t$ is sampled from some distribution $\mathcal{P}_t$ and the distribution $\mathcal{P}_t$ may change over time.

**Theorem 3.** *Under Assumption 2, the Virtual Queue Algorithm of [Neely and Yu, 2017] for any OCOwC problem (denoted by $\pi_2$) produces a decision sequence $\{\boldsymbol{x}_t\}$ such that*

$$\text{Reg}_1(\pi_2, T) \leq O(\sqrt{T}) + \bar{q}W,$$
$$\text{Reg}_2(\pi_2, T) \leq O(d\sqrt{T}).$$

The theorem tells that the non-stationarity when measured properly will not drastically deteriorate the performance of the algorithm for the OCOwC problem as well. Moreover, the non-stationarity will not affect the constraint violation at all. Together with the results for the BwK problem, we argue that the new global non-stationarity measure serves as a proper one for the constrained online learning problems. Note that the upper and lower bounds match up to a logarithmic factor (in a worst-case sense) subject to the non-stationarity measures. The future direction can be to refine the bounds in a more instance-dependent way and to identify useful prior knowledge on the non-stationarity for better algorithm design and analysis.

# 5  Discussions

In this paper, we study the non-stationary setting of the BwK problem. We remark that our formulation is not different from stochastic BwK and adversarial BwK, but it should be viewed as a generalization of both:

- When the underlying distribution $\mathcal{P}_t$ is i.i.d., our formulation degenerates into the stochastic BwK problem.
- Our formulation allows $\mathcal{P}_t$ to be point-mass distributions and also allows it to be chosen adversarially, so it recovers the setting of adversarial BwK. Different from the existing worst-case results on adversarial BwK, our result characterizes a problem-dependent performance that relates the regret with the temporal change of $\mathcal{P}_t$'s.

Two important applications (among others, see Badanidiyuru et al., 2013) are AdWords problem (under pay-by-click and pay-by-conversion), and the pricing problem, where the knapsack constraints capture the bidder's budget or the available inventory. Under such application context, the distribution of the arms' reward and/or resource consumption may change over time; for example, the bidder's bidding policy may change according to their remaining budget, and the underlying market environment may change due to seasonality, day-of-week effect, promotions etc. Both our work and the adversarial BwK aim to capture such violation of the i.i.d. assumption in the stochastic BwK. Speaking of these applications, the stochastic setting is too ideal, while the adversarial setting is too worst-case/conservative; non-stationarity provides a smooth connection between these two ends. The spirit inherits the study of non-stationary environment for unconstrained online learning problem [Besbes et al., 2014, 2015, Cheung et al., 2019, Faury et al., 2021].

Technically, we first make a comparison between the existing results on MAB and on BwK. For the stochastic setting, as we discussed in Appendix D, when the constraints are non-binding, the stochastic BwK's regret bound can recover the regret bound of a corresponding MAB problem. However, for the adversarial setting, the EXP3 algorithm achieves $O(\sqrt{T})$ regret bound for adversarial MAB problem against a static benchmark, while the state-of-the-art adversarial BwK algorithm only achieves an $O(\log T)$ competitiveness ratio against the static benchmark, i.e., even worse than a linear regret. In comparison, we believe the BwK problem in a non-i.i.d. (non-stochastic) environment is pessimistically difficult and far from being resolved. In this light, our work provides a positive result for the problem. We provide in Appendix C1 a detailed discussion and comparison of the benchmarks used in the existing BwK works. Numerical experiments compare the performance of our algorithm with existing BwK algorithms and are presented in Appendix A.

The key for our paper to achieve this result is the proposal of of the new non-stationarity measure, while the algorithm and analysis are largely standard as in the UCB literature. The standard analysis also appears in existing works on non-stationary online learning/optimization [Besbes et al., 2014, 2015, Cheung et al., 2019, Faury et al., 2021] where the techniques more or less follow the paradigm of combining the sliding-window concentration argument with the analysis in a corresponding context of stochastic optimization, MAB, linear bandits, or RL. Our measure is new as all existing works on non-stationarity study unconstrained settings. Our measure is also critical for a constrained setting; we believe its application goes beyond the BwK and OCO problem and it also provides a useful measure for other constrained problems such as constrained MDP and safe RL. Insight-wise, our discussion in Section 2.1 not only validates the criticality of such a measure but also highlights that even if one does not need to perform any learning to the system, the non-stationarity in a constrained problem can still hurt. This intuition is orthogonal to the existing implications of the current works on non-stationarity which mainly focus on remedying the negative effect of nonstationary induced on the learning of the system.

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
