# A    Numerical Experiments

In this section, we examine three algorithms via four numerical examples. The first algorithm is the Sliding Window-UCB (SW-UCB) algorithm presented in our paper. The second algorithm is the naive UCB algorithm without any sliding windows (Agrawal and Devanur, 2014). The third algorithm is LagrangeBwK presented in (Immorlica et al., 2019), which is originally proposed for the adversarial BwK problem. Note that the LagrangeBwK requires an approximation of the static best distribution benchmark. For simplicity, we put the exact value of the benchmark into the algorithm. All the regret performances are reported based on the average over 100 simulation trials.

## A.1    Cumulative Rewards

We first conduct two experiments and plot the cumulative reward of the three algorithms over time.

1. Example 1: One-dimensional $d = 1$. A two-armed instance with *one* resource constraint. $T = 10000$. $B = 5000$. The reward is set to be a constant $\mu_{t,1} = \mu_{t,2} = 0.5$. For the first half time steps $t \leq T/2$, $C_{t,1,1} = 0.5$, $C_{t,1,2} = 1.0$, while for the second half $t > T/2$, $C_{t,1,1} = 1.0$, $C_{t,1,2} = 0.5$. The dynamic optimal benchmark is to play the first arm for the first half and the second arm for the second half. Accordingly, $\text{OPT}(T) = 5000$.

2. Example 2: Two-dimensional $d = 2$. A two-armed instance with *two* resource constraints. $T = 10000$. $B = 5000$. For the first half $t \leq T/2$, we set the reward to be $\mu_{t,1} = \mu_{t,2} = 0.5$. We set $C_{t,1,1} = C_{t,2,2} = 1.0$ and $C_{t,1,2} = C_{t,2,1} = 0$. For the second half $t > T/2$, we force the first arm to be sub-optimal with no reward $\mu_{t,1} = 0$ and maximum consumption $C_{t,1,1} = C_{t,2,1} = 1.0$, while changing the second arm to be optimal with $\mu_{t,2} = 0.5$ and $C_{t,1,2} = C_{t,2,2} = 0.5$. The dynamic optimal benchmark is to play both arms with equal chances for the first half and only the second arm for the second half, yielding $\text{OPT}(T) = 5000$.

For those two examples, the cumulative rewards versus time steps are shown in Figure 1.

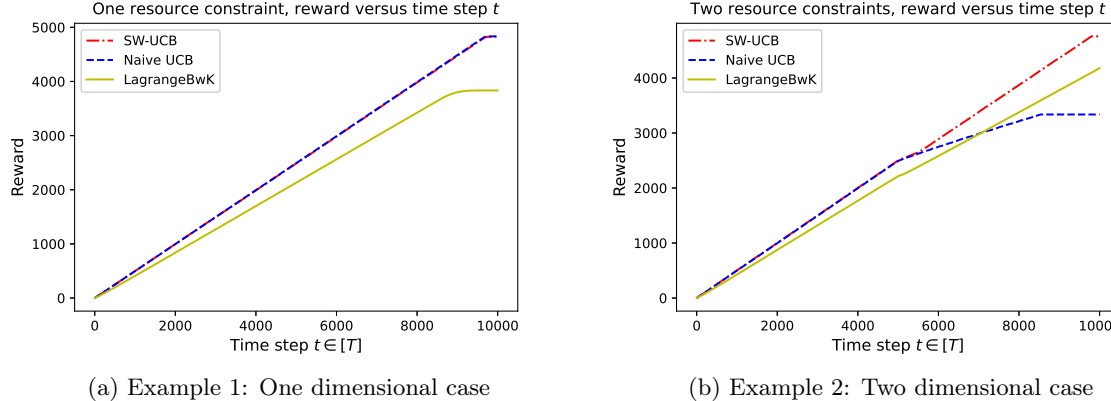

(a) Example 1: One dimensional case              (b) Example 2: Two dimensional case

Figure 1: Cumulative rewards versus time steps.

The performance of the naive UCB algorithm in one-dimensional case (see Figure 1a) is somewhat counter-intuitive: the naive UCB algorithm originally designed for the stochastic setting performs comparable as the SW-UCB and both are better than LagrangeBwK. The reason is that the naive UCB algorithm observes the poor performance of the second arm and ends up with playing the second arm only several hundred times until the second half. So it would not take too long for the naive UCB to rectify its wrong estimate after entering the second half.

But for the two-dimensional case (see Figure 1b), the naive UCB algorithm behaves poorly: it suffers from the abrupt change of the environment and could not adjust its approximation in time. The key difference between this and the one-dimensional setting is that here the optimal distribution for the first half requires playing both arms for sufficiently amount of time rather than simply focusing on one *single* best arm for one-dimensional cases. As a result, the naive UCB algorithm accumulates linearly many observations for both arms during the first half, which significantly affects its performance during the second half. This corresponds to the slope change for the blue curve during the half way.

The algorithm designed specifically for the adversarial BwK, LagrangeBwK, performs slightly worse than our SW-UCB algorithm in both examples. This may be due to the fact that LagrangeBwK acts too conservatively for the cases that are not so *adversarial*.

## A.2 Both $V$ and $W$ Matter

In our upper bound analysis, terms that depend on $V$ and that on $W$ both appear. One may wonder: are both $V$ and $W$ necessary in the analysis? Would it be possible to reduce the terms on $V$ to $W$ or vice versa, reduce $W$ to $V$? In this subsection, we designed two examples to show that *both $V$ and $W$* can make an impact on the performance of BwK algorithms.

The non-stationary BwK problem can be factorized into two sub-problems: identifying the optimal arm distribution with respect to current environment, and finding a resource allocation rule. Larger $V$ makes the first task harder, while larger $W$ creates an obstacle for the second. The following two examples illustrate this intuition.

1. Example 3: Fixed $V$, different $W$'s. A two-armed instance with two resource constraints. $T = 10000$. $B = 2500$. The environment has only one abrupt change point at time $\alpha T$. The local non-stationarity measure $V$ is invariant regardless of the value of $\alpha$, while $W$ depends on $\alpha$. At the first part $t \leq \alpha T$, both arms have a fixed reward $\mu_{t,1} = \mu_{t,2} = 0.5$. As for the consumption, $C_{t,1,1} = C_{t,2,2} = 0.7$, $C_{t,1,2} = C_{t,2,1} = 0.3$. At the second part $t > \alpha T$, $C_{t,1,1} = C_{t,1,2} = C_{t,2,1} = C_{t,2,2} = 1.0$, while the reward $\mu_{t,1} = 0$, $\mu_{t,2} = 0.7$. The dynamic optimal policy is to allocate all the resources to the first part and play both arms with equal chances, which leads to $\mathrm{OPT}(T) = 2500$.

2. Example 4: Fixed $W$, different $V$'s. A two-armed instance with two resource constraints. $T = 10000$. $B = 3125$. The time horizon is still divided into halves. The first half is stationary, while the second half is periodic but with different frequencies. At the first half $t \leq T/2$, both arms are of fixed reward $\mu_{t,1} = \mu_{t,2} = 0.5$. The resource consumption $C_{t,1,1} = C_{t,2,2} = 1.0$, $C_{t,1,2} = C_{t,2,1} = 0$. As for the second half, the first arm now generates no reward $\mu_{t,1} = 0$ but consumes $C_{t,1,1} = C_{t,2,1} = 1.0$. The second arm's reward remains unchanged $\mu_{t,2} = 0.5$, while its consumption $C_{t,1,2} = C_{t,2,2}$ changes across the time horizon according to a piece-wise linear and periodic function ranging from 0 to 1. Here the global non-stationarity measure $W$ is fixed while $V$ varies with respect to the frequency. The dynamic optimal policy is to play both arms with equal chances at the first half but at the second half only the second arm if $C_{t,1,2} = C_{t,2,2} \leq 0.5$. The dynamic optimal benchmark is $\mathrm{OPT}(T) = 3750$.

For above two examples, the algorithm regrets under different $W$ or $V$ are shown in Figure 2.

Both SW-UCB and naive UCB divide the resource evenly and assign each part to each time step, which makes their performance depend on the regret induced by this rule (characterized by $W$ in our analysis) to a very large extent (as is shown in Figure 2a). In comparison, LagrangeBwK is not so sensitive to changing $W$ only, partly due to its Hedge way (see Freund and Schapire (1997)) to allocate resources. For a relatively small $W$ (which means that the problem is not so *adversarial*), one can expect SW-UCB to outperform LagrangeBwK, while for a very large $W$ the adversarial algorithm could be

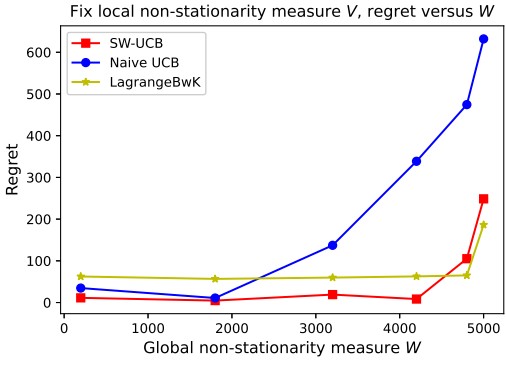
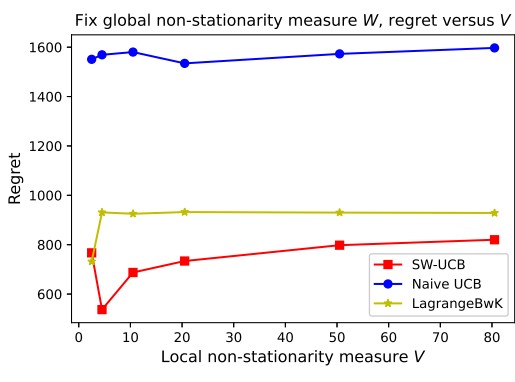

(a) Example 3: Fixed $V$, different $W$'s   (b) Example 4: Fixed $W$, different $V$'s

Figure 2: Regret versus different $W$'s or $V$'s.

better. Note that the performance of naive UCB suffers from the abrupt change of the environment (as is shown in Figure 1b and corresponding analysis).

When $W$ is fixed, the regret varies according to the difficulty in identifying the optimal arm distribution, as is shown in Figure 2b. For those environments that do not vary rapidly, SW-UCB achieves a better regret result, while LagrangeBwK is still insensitive to $V$ due to its adversarial nature. When the environment changes rapidly, SW-UCB fails from precisely learning the environment, while still achieving a better result than LagrangeBwK. We note that naive UCB still performs badly in this two-dimensional case.

# B  Proofs of Section 2 and Section 3

## B.1  Proof of Lemma 1

*Proof.* We first inspect the null arm (say, the $m$-th arm) where $\mu_{t,m} = 0$ and $C_{t,j,m} = 0$. The global DLP must satisfy that

$$\mu_{t,m} - \sum_{j=1}^{d} C_{t,j,m} - (\boldsymbol{\alpha})_t \leq 0,$$

i.e.

$$(\boldsymbol{\alpha})_t \geq 0.$$

The same argument applies to the one-step LP such that $\alpha_t \geq 0$ for all $t = 1, ..., T$.

Note that the reward is upper bounded by 1. Hence,

$$\mathrm{LP}(\{\boldsymbol{\mu}_t\}, \{\boldsymbol{C}_t\}, T) \leq T,$$

$$\mathrm{LP}(\boldsymbol{\mu}_t, \boldsymbol{C}_t) \leq 1, \quad \forall t = 1, \dots, T.$$

Therefore,

$$T\|\boldsymbol{q}^*\|_\infty b \leq T\boldsymbol{b}^\top \boldsymbol{q}^* \leq T\boldsymbol{b}^\top \boldsymbol{q}^* + \sum_{t=1}^{T}(\boldsymbol{\alpha}^*)_t$$
$$= \mathrm{DLP}(\{\boldsymbol{\mu}_t\}, \{\boldsymbol{C}_t\})$$
$$= \mathrm{LP}(\{\boldsymbol{\mu}_t\}, \{\boldsymbol{C}_t\}, T) \leq T,$$

and

$$\|\boldsymbol{q}_t^*\|_\infty b \le \boldsymbol{b}^\top \boldsymbol{q}_t^* \le \boldsymbol{b}^\top \boldsymbol{q}_t^* + \alpha_t^*$$
$$= \mathrm{DLP}(\boldsymbol{\mu}_t, \boldsymbol{C}_t)$$
$$= \mathrm{LP}(\boldsymbol{\mu}_t, \boldsymbol{C}_t) \le 1.$$

Combining above two inequalities together, we have

$$\bar{q}b \le 1.$$

$\square$

## B.2   Proof of Proposition 1

*Proof.* The first inequality is straightforward from the fact that the feasible solutions of single-step $\mathrm{LP}(\boldsymbol{\mu}_t, \boldsymbol{C}_t)$'s yield a feasible solution for the global $\mathrm{LP}(\{\boldsymbol{\mu}_t\}, \{\boldsymbol{C}_t\}, T)$.

For the second inequality, we study the dual problems. By the strong duality of LP, we have

$$\mathrm{DLP}(\{\boldsymbol{\mu}_t\}, \{\boldsymbol{C}_t\}) = \mathrm{LP}(\{\boldsymbol{\mu}_t\}, \{\boldsymbol{C}_t\}, T),$$

$$\mathrm{DLP}(\bar{\boldsymbol{\mu}}, \bar{\boldsymbol{C}}) = \mathrm{LP}(\bar{\boldsymbol{\mu}}, \bar{\boldsymbol{C}}).$$

Denote the dual optimal solution w.r.t. $(\bar{\boldsymbol{\mu}}, \bar{\boldsymbol{C}})$ by $(\bar{\boldsymbol{q}}^*, \bar{\alpha}^*)$. Then

$$\bar{\boldsymbol{\mu}} \le \bar{\boldsymbol{C}}^\top \bar{\boldsymbol{q}}^* + \bar{\alpha}^* \cdot \mathbf{1}_m$$

implies that

$$\boldsymbol{\mu}_t \le \boldsymbol{C}_t^\top \bar{\boldsymbol{q}}^* + \bar{\alpha}^* \cdot \mathbf{1}_m + (\bar{\boldsymbol{C}} - \boldsymbol{C}_t)^\top \bar{\boldsymbol{q}}^* + (\boldsymbol{\mu}_t - \bar{\boldsymbol{\mu}}), \quad \forall t,$$

which induces a feasible solution to the dual program $\mathrm{DLP}(\{\boldsymbol{\mu}_t\}, \{\boldsymbol{C}_t\})$, i.e. $(\bar{\boldsymbol{q}}^*, \boldsymbol{\alpha}')$, where

$$\alpha_t' := \bar{\alpha}^* + \|(\bar{\boldsymbol{C}} - \boldsymbol{C}_t)^\top \bar{\boldsymbol{q}}^* + (\boldsymbol{\mu}_t - \bar{\boldsymbol{\mu}})\|_\infty.$$

Hence,

$$\mathrm{DLP}(\{\boldsymbol{\mu}_t\}, \{\boldsymbol{C}_t\}) - T \cdot \mathrm{DLP}(\bar{\boldsymbol{\mu}}, \bar{\boldsymbol{C}}) \le \sum_{t=1}^{T} \|(\bar{\boldsymbol{C}} - \boldsymbol{C}_t)^\top\|_\infty \|\bar{\boldsymbol{q}}^*\|_\infty + \sum_{t=1}^{T} \|\boldsymbol{\mu}_t - \bar{\boldsymbol{\mu}}\|_\infty$$
$$= \sum_{t=1}^{T} \|\bar{\boldsymbol{C}} - \boldsymbol{C}_t\|_1 \|\bar{\boldsymbol{q}}^*\|_\infty + \sum_{t=1}^{T} \|\boldsymbol{\mu}_t - \bar{\boldsymbol{\mu}}\|_\infty$$
$$\le \bar{q}W_2 + W_1.$$

For the last inequality, similar duality arguments can be made with respect to $T = 1$. Taking a summation, we yield the final inequality as desired. $\square$

## B.3   Proofs of Lemma 2 and Lemma 3

**Lemma 4** (Azuma-Hoeffding's inequality)**.** *Consider a random variable with distribution supported on $[0, 1]$. Denote its expectation as $z$. Let $\bar{Z}$ be the average of $N$ independent samples from this distribution.*

*Then, $\forall \delta > 0$, the following inequality holds with probability at least $1 - \delta$,*

$$|\bar{Z} - z| \leq \sqrt{\frac{1}{2N} \log\left(\frac{2}{\delta}\right)}.$$

*More generally, this result holds if $Z_1, \ldots, Z_N \in [0,1]$ are random variables, $\bar{Z} = \frac{1}{N} \sum_{n=1}^{N} Z_t$, and $z = \frac{1}{N} \sum_{n=1}^{N} \mathbb{E}[Z_n | Z_1, \ldots, Z_{n-1}]$.*

Next, we present a general bound for the normalized empirical mean of the sliding-window estimator:

**Lemma 5.** *For any window size $w$, define the normalized empirical average within window size $w$ of some $Z_{t,i} \in [0,1]$ with mean $z_{t,i}$ for each arm $i$ at time step $t$ as*

$$\hat{Z}_{t,i}^{(w)} := \frac{\sum_{s=1 \vee (t-w)}^{t-1} Z_t \cdot \mathbb{1}\{i_s = i\}}{n_{t,i}^{(w)} + 1},$$

*where $n_{t,i}^{(w)} := \sum_{s=1 \vee (t-w)}^{t-1} \mathbb{1}\{i_s = i\}$ is the number of plays of arm $i$ before time step $t$ within $w$ steps. Then for small $\delta$ such that $\log\left(\frac{2}{\delta}\right) > 2$, the following inequality holds with probability at least $1 - \delta$,*

$$|\hat{Z}_{t,i}^{(w)} - z_{t,i}| \leq \sqrt{\frac{2}{n_{t,i}^{(w)} + 1} \log\left(\frac{2}{\delta}\right)} + \sum_{s=1 \vee (t-w)}^{t-1} |z_{s,i} - z_{s+1,i}|.$$

*Proof.* The result follows from applying Lemma 4 to the empirical mean. For the case when $n_{t,i}^{(w)} = 0$, the result automatically holds. When $n_{t,i}^{(w)} \geq 1$,

$$
\begin{aligned}
|\hat{Z}_{t,i}^{(w)} - z_{t,i}| &\leq \frac{n_{t,i}^{(w)}}{n_{t,i}^{(w)} + 1} \left| \hat{Z}_{t,i}^{(w)} - \frac{\sum_{s=1 \vee (t-w)}^{t-1} z_{s,i} \cdot \mathbb{1}\{i_s = i\}}{n_{t,i}^{(w)}} \right| + \sum_{s=1 \vee (t-w)}^{t-1} \frac{|z_{s,i} - z_{t,i}| \cdot \mathbb{1}\{i_s = i\}}{n_{t,i}^{(w)} + 1} \\
&\quad + \frac{z_{t,i}}{n_{t,i}^{(w)} + 1} \\
&\leq \frac{n_{t,i}^{(w)}}{n_{t,i}^{(w)} + 1} \sqrt{\frac{1}{2 n_{t,i}^{(w)}} \log\left(\frac{2}{\delta}\right)} + \sum_{s=1 \vee (t-w)}^{t-1} \sum_{p=s}^{t-1} \frac{|z_{p,i} - z_{p+1,i}| \cdot \mathbb{1}\{i_s = i\}}{n_{t,i}^{(w)} + 1} \\
&\quad + \sqrt{\frac{1}{2(n_{t,i}^{(w)} + 1)} \log\left(\frac{2}{\delta}\right)} \\
&\leq \sqrt{\frac{2}{n_{t,i}^{(w)} + 1} \log\left(\frac{2}{\delta}\right)} + \sum_{p=1 \vee (t-w)}^{t-1} \frac{\sum_{s=1 \vee (t-w)}^{p} \mathbb{1}\{i_s = i\}}{n_{t,i}^{(w)} + 1} |z_{p,i} - z_{p+1,i}| \\
&\leq \sqrt{\frac{2}{n_{t,i}^{(w)} + 1} \log\left(\frac{2}{\delta}\right)} + \sum_{p=1 \vee (t-w)}^{t-1} |z_{p,i} - z_{p+1,i}|,
\end{aligned}
$$

where the first inequality comes from definition of $\hat{Z}_{t,i}^{(w)}$ and triangular inequality, the second inequality comes from Lemma 4, triangular inequality, and the fact that $z_{t,i} \leq 1 \leq \sqrt{\frac{\log(2/\delta)}{2(n_{t,i}^{(w)} + 1)}}$, the third inequality comes from the fact that $n_{t,i}^{(w)} \leq n_{t,i}^{(w)} + 1$ and rearranging the sum of $p$ and $s$, and the last inequality comes from the fact that $\sum_{s=1 \vee (t-w)}^{p} \mathbb{1}\{i_s = i\} \leq \sum_{s=1 \vee (t-w)}^{t} \mathbb{1}\{i_s = i\} = n_{t,i}^{(w)} \leq n_{t,i}^{(w)} + 1$. $\square$

Using Lemma 5, we can easily derive the proof of Lemma 2:

*Proof.* Replacing $z_{t,i}$ by $\mu_{t,i}$ and $C_{t,j,i}$, $\hat{Z}_{t,i}^{(w)}$ by $\hat{\mu}_{t,i}^{(w_1)}$ and $\hat{C}_{t,j,i}^{(w_2)}$ accordingly in Lemma 5 yields the final result. $\square$

We now start the main proof of Lemma 3.

*Proof.* We will prove the result for reward first. By Lemma 5, with probability at least $1 - \frac{1}{6T}$, $\forall t \leq \min\{\tau, T\}$,

$$
\left| \sum_{s=1}^{t} (\mu_{s,i_s} - \text{UCB}_{s,i_s}(\boldsymbol{\mu}_s)) \right|
$$

$$
\leq 2 \sum_{s=1}^{t} \sqrt{\frac{2}{n_{s,i_s}^{(w_1)} + 1} \log(12mT^3)} + \sum_{s=1}^{t} \sum_{p=1 \vee (t-w_1)}^{s-1} \|\boldsymbol{\mu}_p - \boldsymbol{\mu}_{p+1}\|_\infty
$$

$$
\leq 2 \sum_{s=1}^{t} \sqrt{\frac{2 \log(12mT^3)}{\sum_{p=1 \vee (t-w_1)}^{s-1} \mathbb{1}\{i_p = i_s\} + 1}} + w_1 V_1
$$

$$
\leq 2 \sum_{k=0}^{\lceil t/w_1 \rceil - 1} \sum_{s=kw_1+1}^{(k+1)w_1} \frac{\sqrt{2 \log(12mT^3)}}{\sqrt{\sum_{p=kw_1+1}^{s-1} \mathbb{1}\{i_p = i_s\} + 1}} + w_1 V_1
$$

$$
= 2 \sum_{k=0}^{\lceil t/w_1 \rceil - 1} \sum_{i=1}^{m} \sum_{n=1}^{N_{k,i}^{(w_1)}} \frac{\sqrt{2 \log(12mT^3)}}{\sqrt{n}} + w_1 V_1,
$$

where $N_{k,i}^{(w_1)} := \sum_{p=kw_1+1}^{(k+1)w_1} \mathbb{1}\{i_p = i\}$, $\sum_{i=1}^{m} N_{k,i}^{(w_1)} = w_1$. Here the first inequality comes from Lemma 5, the second inequality comes from the fact that $\sum_{s=1}^{t} \sum_{p=1 \vee (t-w_1)}^{s-1} \|\boldsymbol{\mu}_p - \boldsymbol{\mu}_{p+1}\|_\infty \leq w_1 V_1$, the third comes from cutting time steps into $\lceil t/w_1 \rceil$ periods, and the last comes from rearranging the sum with respect to $i$. Then we have

$$
\left| \sum_{s=1}^{t} (\mu_{s,i_s} - \text{UCB}_{s,i_s}(\boldsymbol{\mu}_s)) \right|
$$

$$
\leq 2 \sum_{k=0}^{\lceil t/w_1 \rceil - 1} \sum_{i=1}^{m} 2\sqrt{2 \log(12mT^3)} \sqrt{N_{k,i}^{(w_1)}} + w_1 V_1
$$

$$
\leq 2 \sum_{k=0}^{\lceil t/w_1 \rceil - 1} 2\sqrt{2 \log(12mT^3)mw_1} + w_1 V_1
$$

$$
\leq 8\sqrt{2 \log(12mT^3)m} \cdot \frac{T}{\sqrt{w_1}} + w_1 V_1, \tag{1}
$$

where the first inequality comes from the fact that $\sum_{n=1}^{N} \frac{1}{\sqrt{n}} \leq 2\sqrt{N}$, the second inequality comes from Cauchy-Schwarz inequality, and the last comes from the fact that $\lceil \frac{t}{w_1} \rceil \leq \frac{2T}{w}$.

Furthermore, applying Lemma 4 to $r_s$ and $\mu_{s,i_s}$, we have that with probability at least $1 - \frac{1}{6mT^2}$, $\forall t \leq \min\{\tau, T\}$,

$$
\left| \sum_{s=1}^{t} (r_s - \mu_{s,i_s}) \right| \leq \sqrt{2T \log(12mT^3)}. \tag{2}
$$

Then we apply Lemma 4 to $\text{UCB}_s(\boldsymbol{\mu}_s)^\top \boldsymbol{x}_s^*$ and $\text{UCB}_{s,i_s}(\boldsymbol{\mu}_s)$ and note that $\text{UCB}_{s,i}(\boldsymbol{\mu}_s) \in [0, 1 + \sqrt{2 \log(12mT^3)}]$. It yields that with probability at least $1 - \frac{1}{6mT^2}$, $\forall t \leq \min\{\tau, T\}$,

$$
\left| \sum_{s=1}^{t} (\text{UCB}_s(\boldsymbol{\mu}_s)^\top \boldsymbol{x}_s^* - \text{UCB}_{s,i_s}(\boldsymbol{\mu}_s)) \right| \leq (1 + \sqrt{2 \log(12mT^3)}) \sqrt{2T \log(12mT^3)}.
$$

Combining all these inequalities (1), (2), (3) together, we have with probability at least $1 - \frac{1}{2T}$, $\forall t \leq$

$\min\{\tau, T\}$,

$$\left| \sum_{s=1}^{t} (r_s - \mathrm{UCB}_s(\boldsymbol{\mu}_s)^\top \boldsymbol{x}_s^*) \right|$$

$$\leq (2 + \sqrt{2\log(12mT^3)})\sqrt{2T\log(12mT^3)} + 8\sqrt{2\log(12mT^3)m} \cdot \frac{T}{\sqrt{w_1}} + w_1 V_1$$

$$\leq 2\sqrt{2\log(12mT^3)} \cdot \sqrt{2T\log(12mT^3)} + 8\sqrt{2\log(12mT^3)m} \cdot \frac{T}{\sqrt{w_1}} + w_1 V_1$$

$$= 4\sqrt{T}\log(12mT^3) + 8\sqrt{2\log(12mT^3)m} \cdot \frac{T}{\sqrt{w_1}} + w_1 V_1, \tag{3}$$

where the first inequality comes from inequalities (1), (2), (3), and the second inequality comes from the fact that $\log(12mT^3) \geq 2$.

As for the resource consumption, by Lemma 5, with probability at least $1 - \frac{1}{6T}$, $\forall t \leq \min\{\tau, T\}$

$$\left| \sum_{s=1}^{t} (C_{s,j,i_s} - \mathrm{LCB}_{s,i_s}(\boldsymbol{C}_{s,j})) \right|$$

$$\leq 2 \sum_{s=1}^{t} \sqrt{\frac{2}{n_{s,i_s}^{(w_2)} + 1} \log(12mdT^3)} + \sum_{s=1}^{t} \sum_{p=1 \vee (t-w_2)}^{s-1} \|\boldsymbol{C}_{p,j} - \boldsymbol{C}_{p+1,j}\|_\infty$$

$$\leq 2 \sum_{s=1}^{t} \sqrt{\frac{2\log(12mdT^3)}{\sum_{p=1 \vee (t-w_2)}^{s-1} \mathbb{1}\{i_p = i_s\} + 1}} + w_2 V_2$$

$$\leq 2 \sum_{k=0}^{\lceil t/w_2 \rceil - 1} \sum_{s=kw_2+1}^{(k+1)w_2} \frac{\sqrt{2\log(12mdT^3)}}{\sqrt{\sum_{p=kw_2+1}^{s-1} \mathbb{1}\{i_p = i_s\} + 1}} + w_2 V_2$$

$$= 2 \sum_{k=0}^{\lceil t/w_2 \rceil - 1} \sum_{i=1}^{m} \sum_{n=1}^{N_{k,i}^{(w_2)}} \frac{\sqrt{2\log(12mdT^3)}}{\sqrt{n}} + w_2 V_2,$$

where $N_{k,i}^{(w_2)} := \sum_{p=kw_2+1}^{(k+1)w_2} \mathbb{1}\{i_p = i\}$, $\sum_{i=1}^{m} N_{k,i}^{(w_2)} = w_2$. Here the first inequality comes from Lemma 5, the second inequality comes from the fact that $\sum_{s=1}^{t} \sum_{p=1 \vee (t-w_1)}^{s-1} \|\boldsymbol{C}_{p,j} - \boldsymbol{C}_{p+1,j}\|_\infty \leq w_2 V_2$, the third comes from cutting time steps into $\lceil t/w_1 \rceil$ periods, and the last comes from rearranging the sum with respect to $i$. Then we have

$$\left| \sum_{s=1}^{t} (C_{s,j,i_s} - \mathrm{LCB}_{s,i_s}(\boldsymbol{C}_{s,j})) \right|$$

$$\leq 2 \sum_{k=0}^{\lceil t/w_2 \rceil - 1} \sum_{i=1}^{m} 2\sqrt{2\log(12mdT^3)}\sqrt{N_{k,i}^{(w_2)}} + w_2 V_2$$

$$\leq 2 \sum_{k=0}^{\lceil t/w_2 \rceil - 1} 2\sqrt{2\log(12mdT^3)mw_2} + w_2 V_2$$

$$\leq 8\sqrt{2\log(12mdT^3)m} \cdot \frac{T}{\sqrt{w_2}} + w_2 V_2, \tag{4}$$

where the first inequality comes from the fact that $\sum_{n=1}^{N} \frac{1}{\sqrt{n}} \leq 2\sqrt{N}$, the second inequality comes from Cauchy-Schwarz inequality, and the last comes from the fact that $\lceil \frac{t}{w_1} \rceil \leq \frac{2T}{w}$.

Similarly, we apply Lemma 4 to $\mathrm{LCB}_s(\boldsymbol{C}_{s,j})^\top \boldsymbol{x}_s^*$ and $\mathrm{LCB}_{s,i_s}(\boldsymbol{C}_{s,j})$ and note that $\mathrm{LCB}_{s,i}(\boldsymbol{C}_{s,j}) \in$

$[-\sqrt{2\log(12mdT^3)}, 1]$. It yields that with probability at least $1 - \frac{1}{6mdT^2}$, $\forall t \leq \min\{\tau, T\}$,

$$\left|\sum_{s=1}^{t}(\text{LCB}_s(\boldsymbol{C}_{s,j})^\top \boldsymbol{x}_s^* - \text{LCB}_{s,i_s}(\boldsymbol{C}_{s,j}))\right| \leq (1 + \sqrt{2\log(12mdT^3)})\sqrt{2T\log(12mdT^3)}. \tag{5}$$

Furthermore, applying Lemma 4 to $c_{s,j}$ and $C_{s,j,i_s}$ induces that with probability at least $1 - \frac{1}{6mdT^2}$, $\forall t \leq \min\{\tau, T\}$,

$$\left|\sum_{s=1}^{t}(c_{s,j} - C_{s,j,i_s})\right| \leq \sqrt{2T\log(6mdT^3)}. \tag{6}$$

Combining all these inequalities (4), (5), (6) together, we have with probability at least $1 - \frac{1}{2T}$, $\forall t \leq \min\{\tau, T\}$,

$$\left|\sum_{s=1}^{t}(c_{s,j} - \text{LCB}_s(\boldsymbol{C}_{s,j})^\top \boldsymbol{x}_s^*)\right|$$

$$\leq (2 + \sqrt{2\log(6mdT^3)})\sqrt{2T\log(6mdT^3)} + 8\sqrt{2\log(12mdT^3)m} \cdot \frac{T}{\sqrt{w_2}} + w_2 V_2$$

$$\leq 4\sqrt{T}\log(6mdT^3) + 8\sqrt{2\log(12mdT^3)m} \cdot \frac{T}{\sqrt{w_2}} + w_2 V_2.$$

$\square$

## B.4   Proof of Corollary 1

*Proof.* Without loss of generality, we only analyze the case that $\tau \leq T$, i.e. the resource constraint is violated before time step $T$. At termination time $\tau$, we have

$$\sum_{t=1}^{\tau} c_{t,j} > bT$$

for some $j \leq d$.

From the fact that $\boldsymbol{x}_t^*$ is a feasible solution to $\text{LP}(\text{UCB}_t(\boldsymbol{\mu}_t), \text{LCB}_t(\boldsymbol{C}_t))$, we have

$$\sum_{t=1}^{\tau} \text{LCB}_t(\boldsymbol{C}_t)\boldsymbol{x}_t^* \leq b\tau.$$

Combining that inequality with Lemma 3, we have with probability at least $1 - \frac{1}{2T}$

$$\sum_{t=1}^{\tau} c_{t,j} \leq b\tau + 4\sqrt{T}\log(12mdT^3) + 14m^{\frac{1}{3}}V_2^{\frac{1}{3}}T^{\frac{2}{3}}\log^{\frac{1}{3}}(12mdT^3) + 8\sqrt{2mT}\sqrt{\log(mdT^3)}.$$

Therefore we have

$$b\tau + 4\sqrt{T}\log(12mdT^3) + 14m^{\frac{1}{3}}V_2^{\frac{1}{3}}T^{\frac{2}{3}}\log^{\frac{1}{3}}(12mdT^3) + 8\sqrt{2mT}\sqrt{\log(mdT^3)} > bT,$$

which yields the final result. $\square$

## B.5 Proof of Theorem 1

*Proof.* From Lemma 2, we know that with probability at least $1 - \frac{1}{3T}$,

$$\sum_{t=1}^{\tau-1} \mathrm{UCB}_t(\boldsymbol{\mu}_t)^\top \boldsymbol{x}_t = \sum_{t=1}^{\tau-1} \mathrm{LP}(\mathrm{UCB}_t(\boldsymbol{\mu}_t), \mathrm{LCB}_t(\boldsymbol{C}_t))$$

$$\geq \sum_{t=1}^{\tau-1} \mathrm{LP}\left(\boldsymbol{\mu}_t - \mathbf{1} \cdot \sum_{s=1\vee(t-w_1)}^{t-1} \|\boldsymbol{\mu}_s - \boldsymbol{\mu}_{s+1}\|_\infty, \ \boldsymbol{C}_t + \sum_{j=1}^{d} \boldsymbol{E}_j \cdot \sum_{s=1\vee(t-w_{2,j})}^{t-1} \|\boldsymbol{C}_{s,j} - \boldsymbol{C}_{s+1,j}\|_\infty\right)$$

$$\geq \sum_{t=1}^{\tau-1} \mathrm{LP}(\boldsymbol{\mu}_t, \boldsymbol{C}_t) - \sum_{t=1}^{\tau-1} \sum_{s=1\vee(t-w_1)}^{t-1} \|\boldsymbol{\mu}_s - \boldsymbol{\mu}_{s+1}\|_\infty - \bar{q}\sum_{t=1}^{\tau-1}\sum_{j=1}^{d}\sum_{s=1\vee(t-w_{2,j})}^{t-1} \|\boldsymbol{C}_{s,j} - \boldsymbol{C}_{s+1,j}\|_\infty$$

$$\geq (\tau-1)\mathrm{LP}(\bar{\boldsymbol{\mu}}, \bar{\boldsymbol{C}}) - (W_1 + \bar{q}W_2) - (w_1 V_1 + \bar{q}dw_2 V_2),$$

where $\boldsymbol{E}_j$ is the matrix that is $\mathbf{1}^\top$ at the $j$-th row while other components all zeros. Here the first inequality comes from Lemma 2, the second inequality comes from the proof of Proposition 1, and the last inequality comes from applying Proposition 1 to $\sum_{t=1}^{\tau-1} \mathrm{LP}(\boldsymbol{\mu}_t, \boldsymbol{C}_t)$.

For Lemma 3, if we select $w_1 = \min\left\{\lceil m^{\frac{1}{3}} V_1^{-\frac{2}{3}} T^{\frac{2}{3}} \log^{\frac{1}{3}}(12mT^3)\rceil, \ T\right\}$, we have $\forall t \leq \min\{\tau, T\}$

$$\left|\sum_{s=1}^{t}(r_s - \mathrm{UCB}_s(\boldsymbol{\mu}_s)^\top \boldsymbol{x}_s^*)\right| \leq 4\sqrt{T}\log(12mT^3) + 14 m^{\frac{1}{3}} V_1^{\frac{1}{3}} T^{\frac{2}{3}} \log^{\frac{1}{3}}(12mT^3) + 8\sqrt{2mT}\sqrt{\log(12mT^3)}.$$

Therefore, by Lemma 3 and Corollary 1, with probability at least $1 - \frac{1}{T}$,

$$\mathrm{LP}(\{\boldsymbol{\mu}_t\}, \{\boldsymbol{C}_t\}, T) - \sum_{t=1}^{\tau-1} r_t$$

$$= (\mathrm{LP}(\{\boldsymbol{\mu}_t\}, \{\boldsymbol{C}_t\}, T) - \sum_{t=1}^{\tau-1} \mathrm{UCB}_t(\boldsymbol{\mu}_t)^\top \boldsymbol{x}_t) + (\sum_{t=1}^{\tau-1} \mathrm{UCB}_t(\boldsymbol{\mu}_t)^\top \boldsymbol{x}_t - \sum_{t=1}^{\tau-1} r_t)$$

$$\leq (4\sqrt{T}\log(12mdT^3) + 14 m^{\frac{1}{3}} V_2^{\frac{1}{3}} T^{\frac{2}{3}} \log^{\frac{1}{3}}(12mdT^3) + 8\sqrt{2mT}\sqrt{\log(12mdT^3)} + 1) \cdot \frac{\mathrm{LP}(\bar{\boldsymbol{\mu}}, \bar{\boldsymbol{C}})}{b}$$

$$\quad + 4\sqrt{T}\log(12mT^3) + 14 m^{\frac{1}{3}} V_1^{\frac{1}{3}} T^{\frac{2}{3}} \log^{\frac{1}{3}}(12mT^3) + 8\sqrt{2mT}\sqrt{\log(12mT^3)}$$

$$\quad + 2(W_1 + \bar{q}W_2) + w_1 V_1 + \bar{q}dw_2 V_2$$

$$= O\left(\frac{1}{b}\sqrt{mT}\log(mdT^3) + m^{\frac{1}{3}} V_1^{\frac{1}{3}} T^{\frac{2}{3}} \log^{\frac{1}{3}}(mT^3) + \frac{1}{b} \cdot m^{\frac{1}{3}} d V_2^{\frac{1}{3}} T^{\frac{2}{3}} \log^{\frac{1}{3}}(mdT^3) + W_1 + \bar{q}W_2\right),$$

where we utilize the fact that $\bar{q}d \leq \frac{1}{b} \cdot d$ by Lemma 1 at the last equality.

Note that OPT is of linear $T$ (in fact, $\mathrm{OPT} \leq \mathrm{LP}(\{\boldsymbol{\mu}_t\}, \{\boldsymbol{C}_t\}, T) \leq T$), which transforms the high probability bound into the expectation bound. $\square$

## B.6 Proof of Theorem 2

*Proof.* The first lower bound follows directly from Besbes et al. (2014). For here, we provide a brief description for completeness. The time horizon is divided into $\lceil \frac{T}{H} \rceil$ periods, where each is of length $H$ except possibly the last one ($H$ to be specified). For each period, the nature selects an arm to be optimal uniformly randomly and independently, which is of mean reward $r^* = \frac{1}{2} + \Delta$, where the other arms are all of $r = \frac{1}{2}$. Then from the information-theoretic arguments of the standard multi-armed bandits problem, if we select $\Delta = \Theta(\sqrt{\frac{m}{H}})$, we must suffer an expected regret of $\Omega(\sqrt{mH})$ at each period for any policy. Here we assume that $H$ is large enough such that $\Delta = \Theta(\sqrt{\frac{m}{H}}) \leq \frac{1}{2}$. Therefore, the total regret is of $\Omega(\sqrt{\frac{m}{H}}T)$, where the local non-stationarity budget $V_1 = \Theta(\Delta\frac{T}{H}) = \Theta(m^{\frac{1}{2}} H^{-\frac{3}{2}} T)$. Note that the

example yields a regret of $\Omega(m^{\frac{1}{3}}V_1^{\frac{1}{3}}T^{\frac{2}{3}})$ by selecting $H = \Theta(m^{\frac{1}{3}}V_1^{-\frac{2}{3}}T^{\frac{2}{3}})$.

For the second lower bound, we can establish based on some modification of the first example. We now assume that each arm is of deterministic reward $r = 1$ and there is only one type of resource. The only difference among the arms is on the resource consumption. To avoid the complication of stopping time, we split the time horizon $T$ into two halves in a way such that the extra consumption of resource at the first half can be deterministically transformed into the reward loss due to limited resource at the second half. Specifically, for the second half, every arm generates a deterministic reward of $r$ and a deterministic consumption $b$. For the first half, the nature divides it in a similar way as the first lower bound example and the goal here is to generate an inevitably excessive resource consumption compared to dynamic optimal policy. There are $\lceil \frac{T}{2H} \rceil$ periods which are of length $H$. Among these periods, the nature uniformly and independently chooses an arm to be optimal. We assume that the optimal arm is of mean consumption $c^* = b$ while the others are of mean consumption $c = b + \Delta = b + \Theta(\sqrt{\frac{m}{H}})$. We can without loss of generality consider only those cases where the resource is not all consumed at the first half (otherwise, the rewards collected will be at least $\frac{1}{2}$OPT smaller than OPT, where the conclusion is automatically fulfilled). By similar information-theoretical arguments, any policy must suffer an expected additional consumption of $\Omega(\sqrt{\frac{m}{H}}T)$ compared to the dynamic optimal policy at the first half, which in turn yields an expected regret of $\Omega(\frac{1}{b}\sqrt{\frac{m}{H}}T)$. By selecting $H = \Theta(m^{\frac{1}{3}}V_2^{-\frac{2}{3}}T^{\frac{2}{3}})$, we construct an example of regret $\Omega(m^{\frac{1}{3}}V_2^{\frac{1}{3}}T^{\frac{2}{3}} \cdot \frac{1}{b})$.

The third lower bound example is constructed based on the motivating example in Section 2.1. We can consider a one-armed bandit problem and divide the time horizon into two halves. There is only one resource type and the total available resource is $B = bT$ with $b < \frac{1}{2}$. At the first half, the arm generates a deterministic reward $r$ and consumes resource $2b$. At the second half, the nature randomly chooses between the following two cases: the situation either becomes better with reward $r + \Delta_1$ and consumption $2b - \Delta_2$ or worse with reward $r - \Delta_1$ and consumption $2b + \Delta_2$. For the first situation, the optimal policy is to reserve the resource as much as possible for the second half whilst for the second situation it is optimal to consume all the resource at the first half. One can choose $r, b$ such that $\frac{r}{b} = \Theta(\bar{q})$. With a similar argument as in Section 2.1, the algorithm will suffer a regret of

$$\Theta(T(\Delta_1 + \frac{r}{b}\Delta_2)) = \Theta(W_1 + \bar{q}W_2)$$

for at least one of the two situations.                                                                    $\square$

# C  Discussions on the Benchmarks and Tightening the Measures

## C.1  Benchmarks used in BwK literature

In the subsection, we provide a thorough discussion on the four benchmarks for the BwK problem. Specifically, our dynamic benchmark is the strongest one in comparison with others, and we are the first one to analyze against this benchmark in a non-stochastic (non-i.i.d.) environment.

$\text{OPT}_{\text{DP}}$: It is defined by an optimal algorithm that utilizes the knowledge of the true underlying distributions and maximizes the expected cumulative reward $E[\sum_{t=1}^{T} r_t]$ subject to the knapsack constraints. This is called as the dynamic optimal benchmark and it is used in the stochastic BwK literature, for both problem-independent bounds (Badanidiyuru et al., 2013; Agrawal and Devanur, 2014), and problem-dependent bounds (Sankararaman and Slivkins, 2021; Li et al., 2021).

$\text{OPT}_{\text{FD}}$: It is called as the fixed distribution benchmark considered in the adversarial BwK problem (Immorlica et al., 2019). It is also defined based on an algorithm that utilizes the knowledge of the true underlying distributions and maximizes the expected cumulative reward. But importantly, the algorithm

is required to play the arms following a fixed (static) distribution throughout the horizon. As mentioned earlier, the dynamic optimal benchmark is more relevant for the practical applications of BwK than this fixed distribution benchmark.

$\text{OPT}_{\text{LP-Dynamic}}$ : It is defined by the optimal value of the following linear program (LP):

$$\text{OPT}_{\text{LP-Dynamic}} \coloneqq \text{LP}\left(\{\mu_t\}, \{C_t\}, T\right) \coloneqq \max_{x_1, \dots, x_T} \sum_{t=1}^{T} \mu_t^\top x_t$$
$$s.t. \sum_{t=1}^{T} C_t x_t \leq B, \quad x_t \in \Delta_m, \ t = 1, \dots, T,$$

and this is the benchmark used in our paper. The LP's inputs $\mu_t$ and $C_t$ are the vector/matrix of the expected reward and resource consumption at time $t$. The decision variables $x_t$ stay within the standard simplex $\Delta_m$ and it can be interpreted as a random arm play distribution for time $t$. The benchmark is also known as deterministic, fluid, or prophet benchmark. It is commonly adopted in the literature for its tractability in analysis than the dynamic benchmark $\text{OPT}_{\text{DP}}$.

$\text{OPT}_{\text{LP-Static}}$ : It is defined by requiring $x_1 = x_2 = \dots = x_T$ in the above LP. This is apparently a weaker benchmark, and it can be viewed as a deterministic upper bound of the $\text{OPT}_{\text{DP}}$.

The following inequality holds

$$\text{OPT}_{\text{FD}} \overset{(1)}{\leq} \text{OPT}_{\text{DP}} \overset{(2)}{\leq} \text{OPT}_{\text{LP-Dynamic}}$$

$$\text{OPT}_{\text{FD}} \overset{(3)}{\leq} \text{OPT}_{\text{LP-Static}} \overset{(4)}{\leq} \text{OPT}_{\text{LP-Dynamic}}.$$

Here (1) and (4) are evident because of the extra requirement of fixed distribution (for (1)) and extra constraint of $x_1 = \dots = x_T$ (for (4)). For (2) and (3), they can be proved by a convexity argument with Jensen's inequality on the realized sample path and the expectation.

We make the following two remarks:

First, when the underlying environment is stochastic (stationary), the expected reward and resource consumption, $\mu_1 = \dots = \mu_T$ and $C_1 = \dots = C_T$. The optimal solution of the LP in defining $\text{OPT}_{\text{LP-Dynamic}}$ automatically satisfies $x_1^* = \dots = x_T^*$. So, for a stochastic environment

$$\text{OPT}_{\text{LP-Static}} = \text{OPT}_{\text{LP-Dynamic}}.$$

The existing literature on stochastic BwK (such as Badanidiyuru et al. (2013); Agrawal and Devanur (2014)) uses this equivalent benchmark to analyze the upper bound of the algorithm regret.

Second, any of these benchmark definition will not restrict it to distributions that do not exhaust budget until T rounds. The LP benchmarks will always upper bound the benchmarks of $\text{OPT}_{\text{FD}}$ and $\text{OPT}_{\text{DP}}$. The LP benchmarks allow early exhaustion as well, because the presence of the null arm allows an play that consume zero resource. This is also reflected by the inequality in the LP's constraints, otherwise if early exhaustion is not allowed, it should be equality in the LP's constraints.

Furthermore, we allow $\mathcal{P}_t$ to be point-mass distributions and allow it to be chosen adversarially. So our non-stationary setting does not conflict with the adversarial setting and it indeed recovers the adversarial BwK as one end of the spectrum. The non-stationarity measures aim to relate the best-achievable algorithm performance with the intensity of adversity of the underlying environment.

## C.2   Tightening the global measures $W_1$ and $W_2$

In this section, we discuss how to improve the global non-stationarity measures $W_1$ and $W_2$. First, we revise the definitions of $W_1$ and $W_2$, and write the non-stationarity measures as functions:

$$W_1(\boldsymbol{\mu}) := \sum_{t=1}^T \|\boldsymbol{\mu}_t - \boldsymbol{\mu}\|_\infty, \quad W_2(\boldsymbol{C}) := \sum_{t=1}^T \|\boldsymbol{C}_t - \boldsymbol{C}\|_1.$$

Specifically, we note that $W_1(\bar{\boldsymbol{\mu}}) = W_1$ and $W_2(\bar{\boldsymbol{C}}) = W_2$.

We note from the proof of Proposition 1 that the following inequalities hold

$$\sum_{t=1}^T \mathrm{LP}(\boldsymbol{\mu}_t, \boldsymbol{C}_t) \le \mathrm{LP}(\{\boldsymbol{\mu}_t\}, \{\boldsymbol{C}_t\}, T) \le T \cdot \mathrm{LP}(\tilde{\boldsymbol{\mu}}, \tilde{\boldsymbol{C}}) + W_1(\tilde{\boldsymbol{\mu}}) + \bar{q} W_2(\tilde{\boldsymbol{C}}) \le \sum_{t=1}^T \mathrm{LP}(\boldsymbol{\mu}_t, \boldsymbol{C}_t) + 2(W_1(\tilde{\boldsymbol{\mu}}) + \bar{q} W_2(\tilde{\boldsymbol{C}}))$$

for any $\tilde{\boldsymbol{\mu}}$ and $\tilde{\boldsymbol{C}}$ if the optimal dual solution $\tilde{\boldsymbol{q}}^*$ of $\mathrm{LP}(\tilde{\boldsymbol{\mu}}, \tilde{\boldsymbol{C}})$ satisfies $\|\tilde{\boldsymbol{q}}^*\|_\infty \le \bar{q}$.

That is, if $\|\tilde{\boldsymbol{q}}^*\|_\infty \le \bar{q}$ is satisfied, the terms of $W_1$ and $W_2$ in Proposition 1 and Theorem 1 can be replaced by $W_1(\tilde{\boldsymbol{\mu}})$ and $W_2(\tilde{\boldsymbol{C}})$, respectively.

Therefore, a natural idea is to refine the two non-stationarity measures based on a combination of $\boldsymbol{\mu}$ and $\boldsymbol{C}$ that minimize the two functions, i.e., to define,

$$W_1^{\min} := \min_{\boldsymbol{\mu}} \sum_{t=1}^T \|\boldsymbol{\mu}_t - \boldsymbol{\mu}\|_\infty,$$

where the optimal solution is denoted by $\boldsymbol{\mu}^*$.

$$W_2^{\min} := \min_{\boldsymbol{C}} \sum_{t=1}^T \|\boldsymbol{C}_t - \boldsymbol{C}\|_1,$$

where the optimal solution is denoted by $\boldsymbol{C}^*$.

**Claim 1.** *The optimal solutions of the two optimization problems above must lie in the convex hull of $\{\boldsymbol{\mu}_t\}$ and $\{\boldsymbol{C}_t\}$, i.e.*

$$\boldsymbol{\mu}^* \in \mathrm{conv}(\{\boldsymbol{\mu}_1, \dots, \boldsymbol{\mu}_T\}), \quad \boldsymbol{C}^* \in \mathrm{conv}(\{\boldsymbol{C}_1, \dots, \boldsymbol{C}_T\}).$$

**Claim 2.** *Any parameter pair $(\tilde{\boldsymbol{\mu}}, \tilde{\boldsymbol{C}})$ that lies in the convex hulls of $\{\boldsymbol{\mu}_t\}$ and $\{\boldsymbol{C}_t\}$ must satisfy the dual price upper bound condition, i.e.*

$$\|\tilde{\boldsymbol{q}}^*\|_\infty \le \max_t \|\boldsymbol{q}_t^*\|_\infty \le \bar{q}, \quad \forall \tilde{\boldsymbol{\mu}} \in \mathrm{conv}(\{\boldsymbol{\mu}_1, \dots, \boldsymbol{\mu}_T\}),\ \tilde{\boldsymbol{C}} \in \mathrm{conv}(\{\boldsymbol{C}_1, \dots, \boldsymbol{C}_T\})$$

*where $\tilde{\boldsymbol{q}}^*$ is the dual optimal solution of the $\mathrm{LP}(\tilde{\boldsymbol{\mu}}, \tilde{\boldsymbol{C}})$.*

**Proposition 3.** *When the above two claims hold, the terms $W_1$ and $W_2$ in the regret bound of Theorem 1 can be replaced by $W_1^{\min}$ and $W_2^{\min}$.*

The proof of the proposition is based on the arguments above, by replacing $W_1$ and $W_2$ in Proposition 1 with $W_1^{\min}$ and $W_2^{\min}$.

Figure 3 provides an illustration of the difference between $W_1$ (or $W_2$) and $W_1^{\min}$ (or $W_2^{\min}$) on a one-armed problem instance. Specifically, consider $\mu_1 = \dots = \mu_k = 1$ and $\mu_{k+1} = \dots = \mu_T = 0$ for some $k$. The optimal choice of $\mu^*$ is to set $\mu^* = 1$ for $k > \frac{T}{2}$ and $\mu^* = 0$ for $k < \frac{T}{2}$, resulting $W_1^{\min} = \min\{k, T-k\}$. Figure 3 plots the two non-stationarity measure against the change point $k$. The result is not contradictory to the lower bound result in that when $k = \frac{T}{2}$, two definitions coincide with the same value.

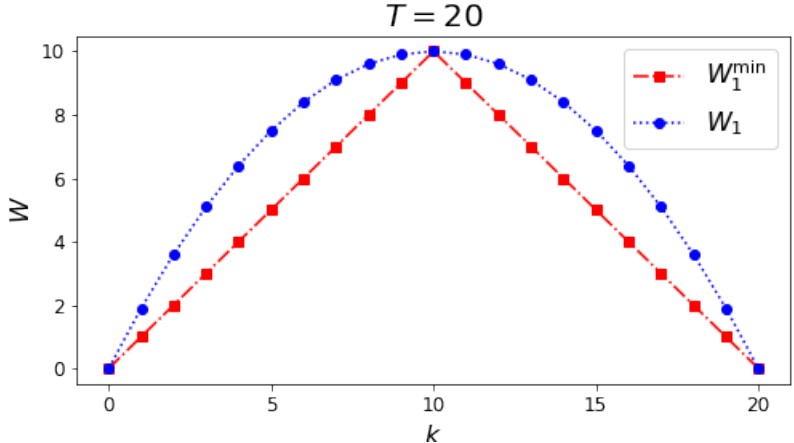

Figure 3: Illustration of $W_1$ and $W_1^{\min}$.

# D  Discussions on the Non-bindingness of the Constraints

As discussed in Section 3, the problem (and the regret bound) of non-stationary BwK can degenerate into the problem of (and the regret bound) non-stationary MAB when all the resource constraints are non-binding. In this section, we elaborate on the statement for two cases: (i) the benchmark LP and all the single-step LPs have all constraints non-binding; (ii) the benchmark LP have all constraints non-binding but some single-step LPs have some binding constraints. Throughout this section, we focus our discussion for the case when $V_1, V_2 > 0$.

## D.1  Non-binding for All LPs

When the benchmark LP and all the single-step LPs have all constraints non-binding, our regret bound will reduce to the regret bound for non-stationary MAB (Besbes et al., 2014). Specifically, when all the LPs have only non-binding constraints, we have

$$\boldsymbol{q}_t^* = \boldsymbol{q}^* = 0.$$

Hence $\bar{q} = 0$ and as a result, the regret bound in Theorem 1 becomes

$$\begin{aligned}
\mathrm{Reg}(\mathrm{Alg}, T) &= O\Big(\frac{1}{b} \cdot \sqrt{mT} \log(mdT^3) + m^{\frac{1}{3}} V_1^{\frac{1}{3}} T^{\frac{2}{3}} \log^{\frac{1}{3}}(mT^3) + \frac{1}{b} \cdot m^{\frac{1}{3}} V_2^{\frac{1}{3}} T^{\frac{2}{3}} \log^{\frac{1}{3}}(mdT^3) + W_1\Big) \\
&= \tilde{O}\Big(m^{\frac{1}{3}} V_1^{\frac{1}{3}} T^{\frac{2}{3}} + \frac{1}{b} \cdot m^{\frac{1}{3}} V_2^{\frac{1}{3}} T^{\frac{2}{3}} + W_1\Big), \quad \forall V_1, V_2 > 0.
\end{aligned}$$

Compared to the regret bound for non-stationary MAB, there are still two additional terms (the terms of $V_2$ and $W_1$). In what follows, we discuss how to remove these two terms in the analysis.

First, it is easy to get rid of the term related to $W_1$. In Proposition 1, we make use of $\mathrm{LP}(\bar{\boldsymbol{\mu}}, \bar{\boldsymbol{C}})$ as a bridge to relates $\sum_{t=1}^{T} \mathrm{LP}(\boldsymbol{\mu}_t, \boldsymbol{C}_t)$ and $\mathrm{LP}(\{\boldsymbol{\mu}_t\}, \{\boldsymbol{C}_t\}, T)$, and this causes the term of $W_1$. When all the constraints are binding for all the LP's, then the following equality naturally holds

$$\sum_{t=1}^{T} \mathrm{LP}(\boldsymbol{\mu}_t, \boldsymbol{C}_t) = \mathrm{LP}(\{\boldsymbol{\mu}_t\}, \{\boldsymbol{C}_t\}, T).$$

As a result, the analysis no longer needs $\mathrm{LP}(\bar{\boldsymbol{\mu}}, \bar{\boldsymbol{C}})$ as a bridge any more, which removes the $W_1$ term.

Second, for the term related to $V_2$, we first point out that this term comes from the bound on

the stopping time in Corollary 1. When the single-step LPs have only non-binding constraints, and the non-bindingness remains stable with right-hand-side being $b' < b$, we can remove this term $V_2$. By remaining stable with $b'$, we mean the single-step LPs have only non-binding constraints if we replace the right-hand-side of the constraints $b$ with $b'$. In this case, for a sufficiently large $T$ such that

$$(b - b')T \geq 4\sqrt{T} \log(12mdT^3) + 9m^{\frac{1}{3}} V_2^{\frac{1}{3}} T^{\frac{2}{3}} \log^{\frac{1}{3}}(12mdT^3),$$

we can apply the arguments in Lemma 3 to show that the stopping time $\tau \geq T$ with high probability. And thus we get rid of the $V_2$ term.

## D.2 Only Benchmark LP Non-binding

When the benchmark LP is non-binding but the single-step LP has binding constraints, we show that the regret bound cannot be reduced to the case of non-stationary MAB. Specifically, consider a one-armed bandit problem instance with an even $T$. There are two types of resources, where each kind is of $\frac{2T}{3}$ budget. For the first half time periods, the arm has reward 1 and consumes 1 unit of resource 1. For the second half, the arm still has reward 1 but consumes 1 unit of resource 2 instead. The global LP is of course nonbinding with $\frac{T}{2} < \frac{2T}{3}$, while the one-step LPs are all binding with $\frac{2}{3} < 1$. The problem instance is in a similar spirit as the motivating example in Section 2.1. In this case, $V_1 = W_1 = 0$ but $\bar{q} > 0$, the terms related to $V_2$ and $W_2$ cannot be removed.

# E Matching the Existing Bound of Stochastic (Stationary) BwK

In this section, we will provide an alternative way to define the upper and lower confidence bounds using a slightly different concentration inequality. As a result, we derive an alternative regret upper bound for the non-stationary BwK problem, which matches the bound (Agrawal and Devanur, 2014) when the environment becomes stationary.

We first state the concentration inequality used in the previous works as a replacement of Lemma 4.

**Lemma 6.** *(Kleinberg et al., 2008; Babaioff et al., 2015; Badanidiyuru et al., 2013; Agrawal and Devanur, 2014). Consider some distribution with values in $[0, 1]$. Denote its expectation by $z$. Let $\bar{Z}$ be the average of $N$ independent samples from this distribution. Then, $\forall \gamma > 0$, the following inequality holds with probability at least $1 - e^{\Omega(\gamma)}$,*

$$|\bar{Z} - z| \leq \text{rad}(\bar{Z}, N) \leq 3\text{rad}(z, N),$$

*where $\text{rad}(a, b) := \sqrt{\frac{\gamma a}{b}} + \frac{\gamma}{b}$. More generally, this result holds if $Z_1, \ldots, Z_N \in [0, 1]$ are random variables, $N\bar{Z} = \sum_{t=1}^{N} Z_t$, and $Nz = \sum_{t=1}^{N} \mathbb{E}[Z_t | Z_1, \ldots, Z_{t-1}]$.*

Then we can revise confidence bounds in Algorithm 1 as follows:

$$\text{UCB}_{t,i}(\boldsymbol{\mu}_t) := \hat{\mu}_{t,i}^{(w_1)} + 2 \cdot \text{rad}(\hat{\mu}_{t,i}^{(w_1)}, n_{t,i}^{(w_1)} + 1),$$

$$\text{LCB}_{t,j,i}(\boldsymbol{C}_t) := \hat{C}_{t,j,i}^{(w_2)} - 2 \cdot \text{rad}(\hat{C}_{t,j,i}^{(w_2)}, n_{t,i}^{(w_2)} + 1),$$

where we will choose the window sizes $w_1, w_2$ according to $V_1, V_2$.

With the new concentration inequality and confidence bounds, Lemma 3 can be replaced by the following two lemmas.

**Lemma 7.** *With probability at least $1 - \frac{1}{T}$, we have*

$$\left|\sum_{t=1}^{T}(r_t - \mathrm{UCB}_t(\boldsymbol{\mu}_t)^\top \boldsymbol{x}_t)\right| \leq O\left(\sqrt{\log(mT^2)\sum_{t=1}^{T}r_t} + \sqrt{\log(mT^2)m} \cdot \frac{T}{\sqrt{w_1}} + w_1 V_1 + \log(mT^2)\right).$$

*If $V_1 > 0$ and we set $w_1 = \Theta(m^{\frac{1}{3}}V_1^{-\frac{2}{3}}T^{\frac{2}{3}}\log^{\frac{1}{3}}(mT^2))$, then*

$$\left|\sum_{t=1}^{T}(r_t - \mathrm{UCB}_t(\boldsymbol{\mu}_t)^\top \boldsymbol{x}_t)\right| = O\left(\sqrt{\log(mT^2)\sum_{t=1}^{T}r_t} + O(m^{\frac{1}{3}}V_1^{\frac{1}{3}}T^{\frac{2}{3}}\log^{\frac{1}{3}}(mT^2))\right) := \beta_1,$$

*with probability at least $1 - \frac{1}{T}$.*

**Lemma 8.** *With probability at least $1 - \frac{1}{T}$, we have $\forall j$,*

$$\left|\sum_{t=1}^{T}(c_{t,j} - \mathrm{LCB}_t(\boldsymbol{C}_{t,j})^\top \boldsymbol{x}_t)\right| \leq O(\sqrt{\log(mdT^2)B_j} + \sqrt{\log(mdT^2)m} \cdot \frac{T}{\sqrt{w_2}} + w_2 V_2 + \log(mdT^2).$$

*If $V_2 > 0$ and we set $w_2 = \Theta(m^{\frac{1}{3}}V_2^{-\frac{2}{3}}T^{\frac{2}{3}}\log^{\frac{1}{3}}(mdT^2))$, then*

$$\left|\sum_{t=1}^{T}(c_{t,j} - \mathrm{LCB}_t(\boldsymbol{C}_{t,j})^\top \boldsymbol{x}_t)\right| = O(\sqrt{\log(mdT^2)B}) + O(m^{\frac{1}{3}}V_2^{\frac{1}{3}}T^{\frac{2}{3}}\log^{\frac{1}{3}}(mdT^2)) := \beta_2,$$

*with probability at least $1 - \frac{1}{T}$.*

Following (Agrawal and Devanur, 2014), one can fulfill the stopping time analysis by shrinking the resource budget in $\mathrm{LP}(\mathrm{UCB}_t(\boldsymbol{\mu}_t), \mathrm{LCB}_t(\boldsymbol{C}_t))$ by $\epsilon$. By choosing an appropriate $\epsilon \geq \frac{\beta_2}{B}$, we can show that the stopping criteria will not be met before $T$ with a high probability, since

$$\sum_{t=1}^{T}c_{t,j} \leq \sum_{t=1}^{T}\mathrm{LCB}_t(\boldsymbol{C}_{t,j}^\top \boldsymbol{x}_t) + \beta_2$$
$$\leq (1 - \epsilon)B + \beta_2$$
$$\leq B.$$

In fact, if we take the stationary case into consideration as well, $\epsilon$ should be slightly larger. Here we use the concentration results in (Agrawal and Devanur, 2014) for the stationary cases directly:

**Lemma 9** (Lemma B.4 in Agrawal and Devanur (2014))**.** *If $V_1 = 0$ and we set $w_1 = T$, then*

$$\left|\sum_{t=1}^{T}(r_t - \mathrm{UCB}_t(\boldsymbol{\mu}_t)^\top \boldsymbol{x}_t)\right| = O(\sqrt{\log(mT^2)m\sum_{t=1}^{T}r_t}) + O(m\log(mT^2)) := \alpha_1,$$

*with probability at least $1 - \frac{1}{T}$.*

**Lemma 10** (Lemma B.5 in Agrawal and Devanur (2014))**.** *If $V_2 = 0$ and we set $w_2 = T$, then*

$$\left|\sum_{t=1}^{T}(c_{t,j} - \mathrm{LCB}_t(\boldsymbol{C}_{t,j})^\top \boldsymbol{x}_t)\right| = O(\sqrt{\log(mdT^2)mB}) + O(m\log(mdT^2)) := \alpha_2, \quad \forall j$$

*with probability at least $1 - \frac{1}{T}$.*

One can choose
$$\epsilon = \frac{\alpha_2 + \beta_2}{B},$$
so that the requirement is met. The shrunken LP will decrease the LP values up to $(1 - \epsilon)$.

Then the final result goes as follows:

**Theorem 4.** *For the non-stationary bandits with knapsacks (NBwK) problem, if $B$ is not too small, i.e.*

$$m^{\frac{1}{3}} V_2^{\frac{1}{3}} T^{\frac{2}{3}} \log^{\frac{1}{3}}(mdT^2) = O(B), \quad \log(mdT^2)m = O(B),$$

*and the dual prices are upper bounded by $\bar{q}$, then with probability at least $1 - \frac{1}{T}$, the regret of the refined sliding-window confidence bound algorithm with $w_1$, $w_2$, and $\epsilon$ selected as suggested (denoted by $\pi_3$) is upper bounded as*

$$\begin{aligned}
\mathrm{Reg}(\pi_3, T) = O( \ & (\sqrt{\frac{m}{B}}\mathrm{OPT}(T) + \sqrt{m\mathrm{OPT}(T)} + m\sqrt{\log(mdT^2)} \ ) \cdot \sqrt{\log(mdT^2)} \\
& + m^{\frac{1}{3}} V_1^{\frac{1}{3}} T^{\frac{2}{3}} \sqrt[3]{\log(mT^2)} + \bar{q} d m^{\frac{1}{3}} V_2^{\frac{1}{3}} T^{\frac{2}{3}} \sqrt[3]{\log(mdT^2)}\cdot \\
& + W_1 + \bar{q}W_2).
\end{aligned}$$

The result meets the upper bounds in (Agrawal and Devanur, 2014) and (Badanidiyuru et al., 2013) up to logarithmic factors when the environment becomes stationary, i.e., $V_1 = V_2 = W_1 = W_2 = 0$. In addition, the regret bound is expressed in terms of $\mathrm{OPT}(T)$. Also note that this upper bound in Theorem 4 does not rely on the linear growth assumption (Assumption 1), but it requires that $B = \Omega(V_2^{\frac{1}{3}} T^{\frac{2}{3}})$ at least for $V_2 > 0$.

# F    Proofs of Section 4

Algorithm 2 describes the Virtual Queue algorithm by Neely and Yu (2017). We first examine its performance with respect to deterministically adversarial constraints $g_{t,i}$'s. We emphasize that in this case, the nature is allowed to choose $g_{t,i}$'s after observing the player's decisions $\{x_1, \ldots, x_{t-1}\}$ as long as the global non-stationarity budget is not violated.

---

**Algorithm 2** Virtual Queue Algorithm for OCOwC Neely and Yu (2017)

---

**Input:** Initial decision $\boldsymbol{x}_0$. Time horizon $T$. Parameters $\beta \leftarrow 1/\sqrt{T}, \alpha \leftarrow 1/T$.
**Output:** Decision sequence $\{\boldsymbol{x}_t\}$.
1: Initialize decision $\boldsymbol{x}_0 \leftarrow \boldsymbol{x}_0$. Initialize virtual queue $Q_i(0) \leftarrow 0, \ Q_i(1) \leftarrow 0$.
2: **while** $1 \leq t \leq T$ **do**
3:    Update virtual queue length if $t \geq 2$:

$$Q_i(t) \leftarrow \max\left\{0, Q_i(t-1) + g_{t-2,i}(\boldsymbol{x}_{t-2}) + \nabla g_{t-2,i}(\boldsymbol{x}_{t-2})^\top (\boldsymbol{x}_{t-1} - \boldsymbol{x}_{t-2})\right\}.$$

4:    Choose $\boldsymbol{x}_t$ as the solution of

$$\operatorname*{arg\,min}_{\boldsymbol{x}\in\mathcal{X}} \left[\beta\nabla f_{t-1}(\boldsymbol{x}_{t-1}) + \sum_{i=1}^{d} Q_i(t)\nabla g_{t-1,j}(\boldsymbol{x}_{t-1})\right]^\top \boldsymbol{x} + \alpha\|\boldsymbol{x} - \boldsymbol{x}_{t-1}\|_2^2.$$

5:    Observe $\nabla f_t(\boldsymbol{x}_t), \nabla g_{t,i}(\boldsymbol{x}_t), i \in [d]$.
6: **end while**

---

We first note that the benchmark taken into consideration in Neely and Yu (2017) is $\mathrm{OPT}'(T)$, which

is based on a more restricted feasible solution set

$$\mathcal{A}' \coloneqq \{\boldsymbol{x} \in \mathcal{X} : \forall t \in [T], \ i \in [d], \ g_{t,i}(\boldsymbol{x}) \leq 0\}$$

compared to the feasible solution set considered in our paper:

$$\mathcal{A} \coloneqq \{\boldsymbol{x} \in \mathcal{X} : \forall i \in [d], \ \sum_{t=1}^{T} g_{t,i}(\boldsymbol{x}) \leq 0\}.$$

Thus our analysis complements the results therein with a more natural benchmark given the global nature of the constraints.

## F.1 Proof of Proposition 2

*Proof.* It follows directly from the fact that $\mathcal{A}' \subset \mathcal{A}$ that

$$\mathrm{OPT}'(T) \geq \mathrm{OPT}(T).$$

We first prove the upper bound. Recall that we denote the primal optimal solution of the standard optimization problem by $\boldsymbol{x}^*$ and that of the restricted optimization problem by $\boldsymbol{x}^{*'}$. Then $\boldsymbol{x}^*$ is a primal feasible solution to the perturbed restricted optimization problem

$$\min_{\boldsymbol{x} \in \mathcal{X}} \sum_{t=1}^{T} f_t(\boldsymbol{x})$$
$$\text{s.t. } g_{t,i}(\boldsymbol{x}) \leq (g_{t,i}(\boldsymbol{x}^*))^+, \quad \forall t \in [T], \ i \in [d].$$

Denote the optimal value of the perturbed problem by $\mathrm{OPT}''(T)$.

Since we have assumed the Slater's condition in Assumption 2, the strong duality holds, and we have

$$\mathrm{OPT}'(T) - \mathrm{OPT}(T) \leq \mathrm{OPT}'(T) - \mathrm{OPT}''(T)$$
$$\leq \bar{q} \sum_{t=1}^{T} \sum_{i=1}^{d} (g_{t,i}(\boldsymbol{x}^*))^+$$
$$= \bar{q} \sum_{i=1}^{d} \sum_{t=1}^{T} (g_{t,i}(\boldsymbol{x}^*))^+$$
$$\leq \bar{q} \sum_{i=1}^{d} (T(\bar{g}_i(\boldsymbol{x}^*))^+ + \sum_{t=1}^{T} \|g_{t,i} - \bar{g}_i\|_\infty)$$
$$= \bar{q} \sum_{t=1}^{T} \sum_{i=1}^{d} \|g_{t,i} - \bar{g}_i\|_\infty$$
$$= \bar{q}W.$$

As for the lower bound, we construct an example based on the idea of the third example given in Theorem 2. The decision set $\mathcal{X}$ is $[0, 1]$. We set the target functions and constraint functions as

$$f_t(x) = -rx,$$

$$g_t(x) = (b + \Delta \mathbb{1}\{t \leq \frac{T}{2}\})x - \frac{1}{2}b.$$

Then the example is now an OCOwC instance with global non-stationarity budgets $W = \Theta(\Delta T)$ and

$\bar{q} = \frac{r}{b}$. The restricted optimal value $\text{OPT}'(T)$ is now

$$\text{OPT}'(T) = -(\frac{T}{4} \cdot \frac{rb}{b+\Delta} + \frac{T}{4} \cdot r),$$

while the standard optimal value $\text{OPT}(T)$ is now

$$\text{OPT}(T) = -\frac{T}{2} \cdot r.$$

If we assume that $\Delta = o(b)$ (which can always be satisfied if we let $b$ to be sufficiently small), then

$$\text{OPT}'(T) - \text{OPT}(T) = \frac{T}{4} \cdot r \cdot \frac{\Delta}{b+\Delta} = \Omega(\bar{q}W).$$

$\square$

## F.2 Proof of Theorem 3

The Virtual Queue algorithm (Algorithm 2) proposed by Neely and Yu (2017) incurs at most $O(\sqrt{T})$ expected regret against the restricted static benchmark $\text{OPT}'(T)$ under Assumption 2. Furthermore, the expected overall constraints violation is bounded by $O(\sqrt{T})$ for each $i \in [d]$ (see detailed analysis in Theorem 1, 3, 4 in Neely and Yu (2017)). While their analysis is against the benchmark $\text{OPT}'(T)$, we here present a result against the stronger benchmark $\text{OPT}(T)$ (Theorem 3).

*Proof.* As is shown in Theorem 1 in Neely and Yu (2017), the Algorithm 2 achieves

$$\sum_{t=1}^{T} f_t(\boldsymbol{X}_t) \leq \text{OPT}'(T) + O(\sqrt{T}).$$

By Proposition 2,

$$\text{OPT}' - \text{OPT}(T) \leq \bar{q}W.$$

Combining above two inequalities together, we have

$$\text{Reg}_1(\pi_3, T) \leq O(\sqrt{T}) + \bar{q}W.$$

As for constraint violation, we shall directly apply Theorem 3 in Neely and Yu (2017) such that

$$\text{Reg}_2(\pi_3, T) = \sum_{i=1}^{d}(\sum_{t=1}^{T} g_{t,i}(\boldsymbol{X}_t))^+ \leq dO(\sqrt{T}).$$

$\square$

## F.3 Extension to the Stochastic Setting

The results in Theorem 3 can be further extended to a stochastic setting where the adversary is oblivious of our decisions. That is, the distributions that govern the random functions $g_{t,i}$ can be chosen adversarially in advance but cannot be adaptively changed according to the decisions $\boldsymbol{x}_t$'s/

We modify the performance measures accordingly as follows

$$\text{Reg}_1(\pi, T) := \mathbb{E}[\sum_{t=1}^{T} f_t(\boldsymbol{X}_t) - \sum_{t=1}^{T} f_t(\boldsymbol{x}^{*'})],$$

$$\text{Reg}_2(\pi, T) := \sum_{i=1}^{d} \mathbb{E}[(\sum_{t=1}^{T} g_{t,i}(\boldsymbol{X}_t))^+],$$

where $\boldsymbol{x}^{*'}$ is the minimizer of $\sum_{t=1}^{T} f_t(\boldsymbol{x})$ on stochastic feasible set

$$\left\{ \boldsymbol{x} \in \mathcal{X} : \mathbb{E}[\sum_{t=1}^{T} g_{t,i}(\boldsymbol{x})] \leq 0 \right\}.$$

We define the certainty equivalent convex programs by

$$\text{OPT}(T) := \min_{\boldsymbol{x} \in \mathcal{X}} \sum_{t=1}^{T} f_t(\boldsymbol{x})$$
$$\text{s.t.} \quad \sum_{t=1}^{T} \mathbb{E}[g_{t,i}(\boldsymbol{x})] \leq 0, \text{ for } i \in [d],$$

and

$$\text{OPT}'(T) := \min_{\boldsymbol{x} \in \mathcal{X}} \sum_{t=1}^{T} f_t(\boldsymbol{x})$$
$$\text{s.t.} \quad \mathbb{E}[g_{t,i}(\boldsymbol{x})] \leq 0, \text{ for } t \in [T], \ i \in [d].$$

Note that Slater's condition in deterministic case can be relaxed to stochastic Slater's condition according to Yu et al. (2017), i.e.

$$\exists \boldsymbol{x}, \text{s.t. } \mathbb{E}[g_{t,i}(\boldsymbol{x})] < 0, \quad \forall t, i.$$

There are some algorithms that are of the same type as the Virtual Queue Algorithm in Neely and Yu (2017) specifically designed for the case with i.i.d. $\boldsymbol{g}_t$'s (see Yu et al. (2017) and Wei et al. (2020)). To obtain similar $O(\sqrt{T})$ regret bound for the stochastic setting, aforementioned papers utilized some similar lemmas (Lemma 6 in Neely and Yu (2017), Lemma 6 in Yu et al. (2017), and Lemma 8 in Wei et al. (2020)) which guarantee that

$$\mathbb{E}\left[\sum_{i=1}^{d} Q_i(t) g_{t-1,i}(\boldsymbol{x}^{*'})\right] \leq 0.$$

For deterministic cases, the above conclusion is reduced to

$$\sum_{i=1}^{d} Q_i(t) g_{t-1,i}(\boldsymbol{x}^{*'}) \leq 0,$$

which automatically holds by the fact that $Q_i(t) \geq 0$ and $g_{t,i}(\boldsymbol{x}^{*'}) \leq 0$. But for stochastic cases, the proof becomes trickier, since we relax the condition $g_{t,i}(\boldsymbol{x}^{*'}) \leq 0$ to $\mathbb{E}[g_{t,i}(\boldsymbol{x}^{*'})] \leq 0$.

In the aforementioned papers, the lemma is proved via factorization of the expectations. Since $Q_i(t)$ is determined by the previous $1 \leq s \leq t-2$ steps' $f_s$'s and $g_{s,i}$'s (note that $\boldsymbol{X}_{t-1}$ is determined by previous $t-2$ steps' functions) while $\boldsymbol{g}_{t-1}$ is independent of previous $t-2$ steps, one can factorize the expectations so that the lemma holds. Specifically, we define two random processes $\{\xi^t\}_{t=1}^{\infty}$ and $\{\gamma^t\}_{t=1}^{\infty}$ such that $f_t(\boldsymbol{x}) = f(\boldsymbol{x}, \xi^t)$ and $\boldsymbol{g}_t(\boldsymbol{x}) = \boldsymbol{g}(\boldsymbol{x}, \gamma^t)$. We define a filtration $\{\mathcal{F}_t : t \geq 0\}$ with $\mathcal{F}_t := \{\xi^\tau, \gamma^\tau\}_{\tau=1}^{t-1}$. Then

taking conditional expectations of $\sum_{i=1}^{d} Q_i(t)g_{t-1,i}(\boldsymbol{x}^{*'})$ yields that

$$\mathbb{E}[\sum_{i=1}^{d} Q_i(t)g_{t-1,i}(\boldsymbol{x}^{*'})|\mathcal{F}_{t-1}] = \sum_{i=1}^{d} Q_i(t)\mathbb{E}[g_{t-1,i}(\boldsymbol{x}^{*'})|\mathcal{F}_{t-1}]$$

$$= \sum_{i=1}^{d} Q_i(t)\mathbb{E}[g_{t-1,i}(\boldsymbol{x}^{*'})] \leq 0$$

where the last equality holds since $\boldsymbol{g}_t$'s are assumed to be i.i.d. in aforementioned papers.

From the above discussions, we can see that the result still holds without the i.i.d. assumption as long as

$$\mathbb{E}[\boldsymbol{g}_t|\mathcal{F}_t] = \mathbb{E}[\boldsymbol{g}_t].$$

Such a requirement is automatically fulfilled when all $\boldsymbol{g}_t$'s are assumed to be distributed independent of previous $\{f_\tau, \boldsymbol{g}_\tau\}_{\tau=1}^{t-1}$'s. Then one can easily derive that the Virtual Queue Algorithm 2 (denoted by $\pi_2$) induces a result of

$$\text{Reg}_1(\pi_2, T) \leq O(\sqrt{T}) + \bar{q}W,$$

$$\text{Reg}_2(\pi_2, T) \leq O(d\sqrt{T}),$$

where

$$W := \sum_{t=1}^{T} \sum_{i=1}^{d} \|\mathbb{E}[g_{t,i}] - \mathbb{E}[\bar{g}_j]\|_\infty,$$

and $\bar{q}$ is the upper bound for the optimal dual solutions of the certainty equivalent convex programs.