# OpenReview forum: "Non-stationary Bandits with Knapsacks"
_NeurIPS.cc/2022/Conference — NeurIPS 2022 Accept_

### Official Review · Reviewer_EkgL · 2022-07-10

**Rating:** 5
**Confidence:** 4
**Soundness:** 2 fair
**Presentation:** 2 fair
**Contribution:** 3 good

**Summary:**

They study the problem of Bandits with Knapsacks where the underlying distribution from which the outcomes are drawn are non-stationary. In this paper, the authors first find a reasonable measure of complexity to measure non-stationarity where we can achieve low-regret. Then they device a sliding-window based algorithm and combine that with UCBBwK from prior work to show that this algorithm achieves low-regret with respect to the new complexity measure they deviced. The paper additionally also proves lower-bound on the best regret achievable.


**Questions:**

he main question I have is around the fact that the defined LP is an upper-bound OPT. First, what type of OPT are we considering here? It seems like we are restricting it to distributions that do not exhaust budget until T rounds. Note in the non-stationary case this coincides with that of the optimal algoritm. However, in the adversarial setting, these are two different benchmarks and one is arguably much weaker than the other. Moreover, this is also weaker than the best fixed distribution that can exhaust budget early (like in Immorlica et al., 2019). The stated fact that the LP is an upper-bound only holds if the OPT considered cannot exhaust budget early. If it can, then this inequality is incorrect. Could you please elaborate and also reflect in the paper that this is the OPT being used? Also some motivation around why this OPT makes sense would be useful to see.


**Limitations:**

Yes, the author's have adequately addressed it.

**Strengths And Weaknesses:**

Originality

- Considers a new setting in the bandits with knapsacks literature not previously considered.
- At the outset, the setting seems that achieving low-regret is not possible (a lower-bound example in Immorlica et al., 2019). However, the authors come up with a different notion of non-stationarity (of course much weaker, but still interesting nonetheless) to bypass this lower-bound and show that low-regret with respect to this complexity measure is possible. Of course, in the lower-bound example case this regret bound reduces to being O(T), but in other more fine-grained settings the regret can be sub-linear.

- In mny opinion, the main contribution of this paper is identifying this notion of complexity. The algorithm idea itself is borrowed from prior works and combined together. The proof requires some work (also see my question below) and then they are able to prove the upper-bound. Its also interesting that they prove a matching (upto log factors) lower bounds.


Quality

- Overall the paper is of high-quality. The considered problem, theory and algorithm are interesting and has new ideas.

Clarity

- The paper is well-written and the main setting, algorithmic idea and related work is explained well. However, I think it would help if the authors described a bit more on the benchmark, what optimum they are comparing against a bit more carefully.

Significance

- The problem setting is a nice mid-ground between the pure stochastic stationary case and the fully pessimistic adversarial case. When the distribution changes over time, but slowly (like in many practical systems) this is a reasonably practical setting. The results are complete in that both upper and lower bounds are proved. The general problem of bandits with knapsacks is well-studied and is interesting to the machine learning community.

The main qualm I have with this paper is that the OPT they have considered seems to be much weaker than prior works. I wonder, if the authors can provide more motivation on why this OPT makes sense.

---

> ### Author Response · Authors · 2022-08-02
> **Discussion on Benchmarks**
>
> We thank the reviewer for the helpful comments, and we are sorry for the caused confusion in explaining the various benchmarks. In the following, we elaborate the four benchmarks for the BwK problem. Specifically, our dynamic benchmark is the strongest one in comparison with others, and we are the first one to analyze against this benchmark in a non-stochastic (non-i.i.d.) environment.
>
> $OPT_{DP}$: It is defined by an optimal algorithm that utilizes the knowledge of the true underlying distributions and maximizes the expected cumulative reward $E[\sum_{t=1}^T r_t]$ subject to the knapsack constraints. This is called as the dynamic optimal benchmark and it is used in the stochastic BwK literature, for both problem-independent bounds (Badanidiyuru et al., 2013; Agrawal and Devanur, 2014), and problem-dependent bounds (Sankararaman
> and Slivkins, 2021; Li et al., 2021).
>
> ${OPT}_{{FD}}$: It is called as the fixed distribution benchmark considered in the adversarial BwK problem (Immorlica et al., 2019). It is also defined based on an algorithm that utilizes the knowledge of the true underlying distributions and maximizes the expected cumulative reward. But importantly, the algorithm is required to play the arms following a fixed (static) distribution throughout the horizon. We refer to our response to Reviewer jSi5 for practical settings where the dynamic optimal benchmark is more relevant than this fixed distribution benchmark.
>
> $OPT_{LP-Dynamic}$: It is defined by the optimal value of the following linear program (LP):
>
>  $OPT_{LP-Dynamic} \coloneqq \mathrm{LP}\left(\{ \mu_t\}, \{ C_t\}, T\right) \ \coloneqq  \max_{ x_1,\dots, x_T} \ \sum_{t=1}^T  \mu_t^\top  x_t$
>
>   ${s.t. } \sum_{t=1}^T  C_t  x_t \le  B,\quad  x_t \in \Delta_m,\ t = 1,\dots, T$
>
> and this is the benchmark used in our paper. The LP's inputs $\mu_{t}$ and $C_t$ are the vector/matrix of the expected reward and resource consumption at time $t$. The decision variables $x_t$ stay within the standard simplex $\Delta_m$ and it can be interpreted as a random arm play distribution for time $t$. The benchmark is also known as deterministic, fluid, or prophet benchmark. It is commonly adopted in the literature for its tractability in analysis rather than the dynamic benchmark $OPT_{{DP}}$.
>
> $OPT_{{LP-Static}}:$ It is defined by requiring $x_1=x_2=...=x_T$ in the above LP. This is apparently a weaker benchmark, and it can be viewed as a deterministic upper bound of the $OPT_{DP}$.
>
> The following inequality holds
>
> $OPT_{FD} \overset{(1)}{\le} OPT_{DP}  \overset{(2)}{\le }OPT_{LP-Dynamic} $
>
> $OPT_{FD}\overset{(3)}{\le} OPT_{LP-Static} \overset{(4)}{\le} OPT_{LP-Dynamic}.$
>
> Here (1) and (4) are evident because of the extra requirement of fixed distribution (for (1)) and extra constraint of $x_1=...=x_T$ (for (4)). For (2) and (3), they can be proved by a convexity argument with Jensen's inequality on the realized sample path and the expectation.
>
> We make the following remarks:
>
> First, when the underlying environment is stochastic (stationary), the expected reward and resource consumption, $\mu_1=...=\mu_T$ and $C_1=...=C_T.$ The optimal solution of the LP in defining ${OPT}_{{LP-Dynamic}}$ automatically satisfies $x_1^* =...=x_T^*.$ So, for a stochastic environment
>
> $OPT_{LP-Static}=OPT_{LP-Dynamic}.$
>
> The existing literature on stochastic BwK (such as Badanidiyuru et al., 2013; Agrawal and Devanur, 2014) uses this equivalent benchmark to analyze the upper bound of the algorithm regret.
>
> Second, any of these benchmark definitions will not "restrict it to distributions that do not exhaust budget until T rounds". The LP benchmarks will always upper bound the benchmarks of $OPT_{FD}$ and $OPT_{DP}.$ The LP benchmarks allow early exhaustion as well, because the presence of the null arm allows a play that consumes zero resource. This is also reflected by the "$\le$" in the LP's constraints, otherwise if early exhaustion is not allowed, it should be "=" in the LP's constraints.
>
> Furthermore, we allow $\mathcal{P}_t$ to be point-mass distributions and allow it to be chosen adversarially. So our non-stationary setting does not conflict with the adversarial setting and it indeed recovers the adversarial BwK as one end of the spectrum. The non-stationarity measures aim to relate the best-achievable algorithm performance with the intensity of adversity of the underlying environment.
>
> We discuss the practical justifications for considering such benchmarks in the response to Reviewer jSi5. Also, we update the supplementary material with some numerical experiments to illustrate our algorithm performance against the existing algorithms. We believe the advantage of our algorithm is robust to any possible underlying environment.
>
> We look forward to your further feedback and will provide our clarification timely.

---

### Official Review · Reviewer_3Kci · 2022-07-10

**Rating:** 6
**Confidence:** 3
**Soundness:** 3 good
**Presentation:** 3 good
**Contribution:** 3 good

**Summary:**

In this paper, the authors studied bandits with knapsacks in a non-stationary environment, which differs from existing literature that studied this problem under either the stationary or adversarial settings. The authors characterized a notion called global non-stationary budget, which is different from the notion of variation budget and required due to the presence of the knapsack constraints. They presented a sliding-window UCB algorithm for the non-stationary BwK, established both regret upper and lower bounds, and showed that the algorithm is near-optimal. Finally, the authors also discussed how the non-stationary measure can also be applied in the problem of online convex optimization with constraints.


**Questions:**

1. Is the linear growth assumption only used for introducing the single-step LP? What happens when we do not have this assumption?
2. Could you provide some intuition on why we have linear dependency on $d$ in the third term of the regret bound on Theorem 1? Any thoughts on why $\log{d}$ might be attainable?
3. Could you provide some more intuitions/details about how $W_1^{\min}$ and $W_2^{\min}$ differ from $W_1$ and $W_2$? Why does it make no change to the analysis?


**Limitations:**

None.

**Strengths And Weaknesses:**

Strengths:
1. The notion of global non-stationary budget is novel and appears crucial to the problem with knapsack constraints. The authors also provided good intuitions on why the common variation budget does not fit the constrained setting in Section 2.1.
2. The sliding-window algorithm presented in Section 3 is shown to attain a near-optimal regret bound based on the results shown in Theorem 1 & 2. The theoretical results here are well explained and appear solid. The authors also provided detailed explanations on how these results compared to previous works.
3. The authors also applied the non-stationary measure beyond the bandits with knapsack framework and showed that it’s also a proper measure for constrained convex optimization.

Weaknesses:
1. More motivations should be provided for why the authors wished to consider such a setting. Is it simply combining the works from (i) bandits with knapsack and (ii) non-stationary bandits? What kind of real-world problems can we model using such a setting?
2. While the results presented in this paper appear theoretically sound, it is unclear how the proposed algorithms perform in practice. What are the realistic implications of the global non-stationary budget? Some numerical experiments could be helpful in illustrating the performance of the algorithm under different variation budgets $W_1$ and $W_2$.

---

> ### Author Response · Authors · 2022-08-02
> **Motivation, experiment, and technical discussions**
>
> We thank the reviewer for all the comments.
>
> For our modeling motivation, we first remark that our formulation is not different (or in conflict) from stochastic BwK and adversarial BwK, but it should be viewed as a generalization of both:
> -	When the underlying distribution $\mathcal{P}_t$ is i.i.d., our formulation degenerates into the stochastic BwK problem.
> -	Our formulation allows $\mathcal{P}_t$ to be point-mass distributions and also allows it to be chosen adversarially, so it recovers the setting of adversarial BwK.
>
> Two important applications (among others, see Badanidiyuru et al., 2013) are AdWords problem (under pay-by-click and pay-by-conversion), and the pricing problem, where the knapsack constraints capture the bidder’s budget or the available inventory. Under such application contexts, the distribution of the arms’ reward and/or resource consumption may change over time; for example, the bidder’s bidding policy may change according to their remaining budget, and the underlying market environment may change due to seasonality, day-of-week effect, promotions etc. In this light, the stochastic setting is too ideal, while the adversarial setting is too worst-case; non-stationarity provides a smooth connection between these two ends.
>
> Technically, we consider a dynamic benchmark while the adversarial BwK results consider a static benchmark (technical discussion referred to our response for Reviewer EkgL). The dynamic benchmark is more relevant for the applications such as pricing and Adwords. In such contexts, one hopes to have a performance guarantee against a benchmark where the airlines/hotels/ski-resorts can price differently for different dates, and the Ads platforms can perform a different allocation rule based on the remaining budgets of the Ads bidders (more discussion referred to our response to Reviewer jSi5). In addition, we achieve a sublinear regret bound (provided W_1 and W_2 not scaling linearly with T) while the existing adversarial BwK results only obtain O(log T) competitiveness ratio.
>
> We conduct more numerical experiments and update our supplementary material. We really appreciate if you have time to read them. The numerical experiments show that the stochastic BwK algorithm can fail drastically even when the underlying distribution changes only once over time. Our algorithm, though not much novelty in its design idea, performs consistently better than both the stochastic BwK algorithm and the adversarial BwK algorithm.
>
> “linear growth assumption”
> We remark that the linear growth assumption is just for notation simplicity. One can replace every $b$ with $B/T$ and will not change the analysis at all. That’s how we recover the previous regret bound in Appendix D where the previous works make no such assumption. We chose this $b$ notation for two reasons: 1. It gives a natural weight for the non-stationarity of the constraints in Theorem 1, in other words, how we discount the effect of constraints into the reward objective. 2. It provides an upper bound on the dual price (Lemma 1), i.e., maximum amount of reward achievable by consuming each unit of the resource.
>
> “dependency on $d$”
> The linear dependency on the number of constraints $d$ comes from the first block of inequalities in the proof of Theorem 1 (Appendix B.5). Essentially, it is due to the definition of $V_2 = \max V_{2,j}.$ If we change the definition to $V_2 = \sum_{j} V_{2,j},$ this factor of $d$ will be removed. Also, if there is additional structure to sharpen the above-mentioned inequalities, the factor of $d$ can also be improved. We will include more discussion in the future version of our paper.
>
> “$W_1^{\min}$ and $W_2^{\min}$”
>
> Consider the following problem: there are T points spreading through a d-dimensional space, where we want to find a point that minimizes the sum of $L_1$ distance from those points. The arithmetic average is not the exact solution. Instead, the minimum is obtained via a linear program, which is just the case of $W_1^{min}$ and $W_2^{min}$. We do not assume the knowledge of the underlying distributions, so we cannot define such a linear program precisely. But one can regard $W_1$ and $W_2$ as a same-order approximation of $W_1^{min}$ and $W_2^{min}$. To see that, we note that the minimization of $L_1$ norm can be done component-wise, and the 1-dimensional case is straightforward.
>
> There are only three parts in our analysis that contain $W_1$ or $W_2$: Proposition 1 (the second and third inequalities), Theorem 1 (where Proposition 1 has been applied twice), and Theorem 2 (the third example). Proposition 1 remains all the same as long as $\bar{\mu}$ and $\bar{C}$ are replaced by the corresponding $\mu$ and $C$ of $W_1^{min}$ and $W_2^{\min}$ and their dual optimal solutions are also replaced accordingly. Theorem 1 is related to $W$ only via Proposition 1, meaning that it will keep unchanged. The third example we construct for Theorem 2 itself satisfies $W_1=W_1^{min}, W_2=W_2^{min}$.

---

> > ### Comment · Reviewer_3Kci · 2022-08-09
> > **Thank you for the response!**
> >
> > I thank the authors for putting together the detailed response above and they have also addressed my questions about the technical results. Overall I think the paper is well written and the technical results are well organized and presented. In terms of further improvements, it would be great if the authors can add an extended discussion about how they motivate the current setting (I like the two applications that the authors discussed in the response, and it would be good if they can also discuss what are the real-world implications of local/global non-stationary budgets under these settings). I also agree with Reviewer JHmo that the contributions of the current work can be further highlighted, as the technical results appear similar to prior works. (I think the introduction of the non-stationary measure is novel, but I wonder if there are any new ideas for the algorithmic design?) It'd be great if the authors can incorporate their extended discussion of the contribution into the main text. Due to these reasons, I will keep my original score unchanged.

---

> > > ### Author Response · Authors · 2022-08-09
> > > **thank you for the feedback**
> > >
> > > Thanks very much for your comments. We appreciate your suggestions on the exposition of our paper and we will improve accordingly in future versions. We'd also like to take this opportunity to thank all the reviewers for the helpful feedbacks in improving our paper.
> > >
> > > We don’t want to bother you, other reviewers, and the ACs  to read a long response, so we will make a few short comments as follows:
> > >
> > > Interpretation of local/global non-stationarity measures in application contexts:
> > >
> > > - the local non-stationarity measure captures the learnability of the environment: Generally, the best one can do to estimate the environment is to use recent observations, and the local measure captures the effectiveness of such estimators.
> > > - the global non-stationarity measure captures the possibility of an effective resource allocation scheme: This is a special point for the constrained online learning problem where the decision maker needs to allocate the resource in a temporal manner against future uncertainty.
> > >
> > > The sliding-window design is all around in the literature on non-stationary online learning. Our contribution to the algorithmic side is more on a new analysis of the sliding-window algorithm with the proposed non-stationarity measure and to provide a positive answer for the BwK problem (not "so" difficult to permit only competitiveness ratio). We make the following two additional remarks:
> > >
> > > 1. Our algorithm doesn't require the knowledge of the global measure $W_1$ and $W_2$. From the lower bound and the motivating example in Section 2.1, even with the knowledge of these two, one cannot further improve the regret. This is in contrast to the existing works using variation budget (our local measure) which all require knowing the budget. Our algorithm still requires the knowledge of the local measure $V_1$ and $V_2$ and this requirement is inherited from this line of literature.
> > > 2. After reading your feedback, we did a quick check on another setting of non-stationarity: the piecewise-stationary setting (for example, Cao et al. 2019). We find that our global measure is also compatible with the change point measure (the number of times the underlying distribution $P_t$ changes over time). Our algorithm can be adjusted accordingly with a subroutine for change point detection, and in this setting, it doesn't require any prior knowledge of the environment which becomes entirely distinct from the variation budget literature. We are more than happy to include this in the later version of our paper.
> > >
> > > Ref: Nearly optimal adaptive procedure with change detection for piecewise-stationary bandit, Cao et al. 2019

---

### Official Review · Reviewer_JHmo · 2022-07-13

**Rating:** 4
**Confidence:** 3
**Soundness:** 3 good
**Presentation:** 2 fair
**Contribution:** 2 fair

**Summary:**


The authors studied the problem of bandits with knapsacks (BwK) in a non-stationary environment. In BwK problem the objective is to maximize the cumulative reward over a finite horizon subject to some knapsack constraints on the resources. Their first result shows that the traditional notion of variation budget is insufficient to characterize the non-stationarity of the BwK problem for a sublinear regret due to the presence of the constraints. Following, the authors propose a new notion of global non-stationarity measure and establish upper and lower bounds for the problem. The results are based on a primal-dual analysis of the underlying linear programs which highlights the interplay between the constraints and the non-stationarity. Finally, the paper also extends to the setting of online convex optimization with constraints and analyze new regret bounds.

**Questions:**

Is there any new idea introduced in this paper? I agree this being the first work on non-stationary-BwK, however the problem formulation seems to be an extention from that of MAB literature and so are the algorithmic ideas + analysis techniques. For example, the sliding window UCB idea is popularly studied in non-stationary bandit literature. The subsequent regret analysis and the nature of the bounds also look very similar to that of MAB (e.g. O(V^1/3 T^2/3) regret etc.). The measure of non-stationarity in Non-stationary BwK (or OCOwC) are new but they are most intuitive ones and possibly can not be considered as enough contribution for this venue.

**Limitations:**

The lack of technical novelty in the problem formulation, as well as the solution approach (both of which seem like an aggregation of ideas from MAB literature), is the key limitation of the paper.

**Strengths And Weaknesses:**


Strength:
- Strong theoretical results with matching lower bounds guarantees for some cases
- Motivating applications


Weakness:
- Technical novelty in problem formulation as well as algorithmic ideas

---

> ### Author Response · Authors · 2022-08-02
> **result contribution and technical contribution**
>
> We thank the reviewer for the comments.
> As the reviewer mentioned, we’d like to start our discussion with a comparison between the BwK problem and the standard MAB problem. For the stochastic setting, as we discussed in Appendix D, when the constraints are non-binding, the stochastic BwK’s regret bound can recover the regret bound of a corresponding MAB problem. Yet for the adversarial setting, the EXP3 algorithm achieves $O(\sqrt{T})$ regret bound for adversarial MAB problem against a static benchmark, while the state-of-the-art adversarial BwK algorithm only achieves an $O(\log T)$ competitiveness ratio against the static benchmark, i.e., worse than a linear regret. In comparison, we believe the BwK problem in a non-i.i.d. (non-stochastic) environment is pessimistically difficult and far from being resolved. In this light, our work provides a positive result for the problem.
>
> Contributions of our results:
> Necessity of a non-i.i.d./beyond-stochastic environment:
> The applications of BwK problem are quite different from the applications of the MAB problem. We refer to our response to Reviewer jSi5 for a detailed discussion. In such application contexts, the stochastic setting is too ideal, and on the other hand, the adversarial setting is too worst-case. Specifically, the underlying marketing environment/customers will not really act in a worst-case adversarial manner against the decision maker. In this light, our formulation umbrellas the two existing settings (stochastic and adversarial) as the two ends of the spectrum.
>
> Necessity of a dynamic benchmark:
> The existing works on adversarial BwK consider a static benchmark, while our work is the first result that competes against the stronger dynamic benchmark. We refer the math details of these two benchmarks to our response to Reviewer EkgL. The dynamic benchmark is more relevant in the application context (See our response to Reviewer jSi5) of the BwK problem. For example, consider a pricing problem for a hotel, then we hope to develop an algorithm that admits a performance guarantee against the best dynamic pricing policy (which allows different prices over different dates/time periods), instead of an algorithm that only has a guarantee against the best static pricing policy (best fixed price across all dates).
>
> Our technical contributions:
> - The proposal of the measure: In this paper, we propose a new non-stationarity measure. In the literature on non-stationary online learning/optimization (Besbes et al., 2014, 2015; Cheung et al., 2019; Faury et al., 2021), the techniques more or less follow the paradigm of combining the sliding-window concentration argument with the analysis in a corresponding context of stochastic optimization, MAB, linear bandits, or RL. We believe the impactfulness of work along this line is mainly evaluated based on (i) the criticalness of the proposed non-stationarity measure, and (ii) the algorithmic and analytical insights the measure brings to the table. Our measure is new as all existing works on non-stationarity study unconstrained settings. Our measure is also critical for a constrained setting; we believe its implication  goes beyond the BwK and OCO problem and it provides a useful (possible unavoidable) measure for other constrained problems such as constrained MDP and safe RL. Insight-wise, our discussion in Section 2.1 not only validates the criticalness of such a measure but also highlights that even if one doesn’t need to perform any learning to the system, the non-stationarity in a constrained problem can still hurt. This intuition is orthogonal to the existing implications of the current works on non-stationarity which mainly focus on remedying the negative effect of nonstationary induced on the learning of the system.
> - Dual-based analysis: A brute-force application of the sliding window argument will fail for the analysis. To see the reason, consider the example in Section 2.1, if one simply applies the concentration argument to bound the stopping time (depletion time) of the resources, it will give a $T-\tau = \Omega(T)$ and correspondingly a linear regret for the problem. Instead, we take a primal-dual approach to relate the consumption of the resources with the collected reward. This is made concrete through Lemma 1, Proposition 1, and Theorem 1. The introduction of the dual optimal solution upper bound $\bar{q}$ is new to the literature of BwK and critical in the analysis, which can also be verified by its appearance in both upper and lower bounds of the problem.
> - We provide a thorough analysis of the reduction of BwK in a non-stationary (non-i.i.d.) environment to (i) a stochastic BwK problem (after Theorem 1 and Appendix D); (ii) a non-stationary MAB problem (Appendix C). We remove the redundant $(1-\epsilon)$ shrinkage design in the algorithm of (Agrawal and Devanur, 2014), and clean the analysis accordingly. We are also the first to allow a non-i.i.d. constraint for OCO problems.

---

### Official Review · Reviewer_jSi5 · 2022-07-21

**Rating:** 5
**Confidence:** 3
**Soundness:** 2 fair
**Presentation:** 3 good
**Contribution:** 2 fair

**Summary:**

The paper extends the problem of multiarmed bandits with knapsack to the situation where the underlying environment is allowed to vary. Further, the paper extends the sliding window UCB algorithm to be applied to this setting. The authors show lower bounds for this problem and also show how the extension of the sliding window UCB algorithm can achieve near optimal worst-case regret. Finally, they extend their methods to general online convex optimization problems with constraints.

**Questions:**

1) I am  unclear of the usefulness/applicability of such a model. Stochastisticity and Adversity make sense to me, but can the authors give a concrete practical situation (other than for theoretical completion) where such a setting indeed makes sense.

2) Can the authors supply a simulation result where standard SOTA algorithms designed stochastic knapsack problem fail when applied to the setting proposed, while the algorithm proposed overcomes this.



**Limitations:**

While the paper does not have a "Discussion" or "Conclusion" section (due to paucity of space probably), the authors provide some hints on the future directions and the limitations of their work.

**Strengths And Weaknesses:**

+ I really like the presentation of the paper, its very well-written especially Sec 2 and 3.
+ The theoretical results on the regret bounds, both the upper and lower bounds, seem technically sound


- While doing a good job on explaining things, still I am a little unsure about the motivation of the problem itself, i.e., how and when such a setting is applicable given that one already has algorithms for the stochastic and adversarial settings. Let me know if I missed something.
- The paper can benefit further with some simulations comparing how standard SOTA algorithm for stochastic knapsack problem can fail when applied to such settings, while the modification proposed can overcome such an issue.

---

> ### Author Response · Authors · 2022-08-02
> **Motivation and numerical experiments**
>
> We thank the reviewer for the comments.
>
> Application/motivation for our non-stationary formulation:
>
> Our formulation is not different from stochastic BwK and adversarial BwK, but it should be viewed as a generalization of both:
> - When the underlying distribution $\mathcal{P}_t$ is i.i.d., our formulation degenerates into the stochastic BwK problem.
> - Our formulation allows $\mathcal{P}_t$ to be point-mass distributions and also allows it to be chosen adversarially, so it recovers the setting of adversarial BwK. Different from the existing worst-case results on adversarial BwK, our result characterizes a problem-dependent performance that relates the regret with the temporal change of $\mathcal{P}_t$’s.
>
> Two important applications (among others, see Badanidiyuru et al., 2013) are AdWords problem (under pay-by-click and pay-by-conversion), and the pricing problem, where the knapsack constraints capture the bidder’s budget or the available inventory. Under such application context, the distribution of the arms’ reward and/or resource consumption may change over time; for example, the bidder’s bidding policy may change according to their remaining budget, and the underlying market environment may change due to seasonality, day-of-week effect, promotions etc. Both our work and the adversarial BwK aim to capture such violation of the i.i.d. assumption in the stochastic BwK. Speaking of these applications, the stochastic setting is too ideal, while the adversarial setting is too worst-case; non-stationarity provides a smooth connection between these two ends. The spirit inherits the study of non-stationary environment for unconstrained online learning problem (Besbes et al., 2014, 2015; Cheung et al., 2019; Faury et al., 2021).
>
> Now we elaborate how our results contribute upon the existing results of adversarial BwK:
> - The adversarial BwK considers a static benchmark while our paper considers a stronger dynamic benchmark in regret analysis. The static benchmark is defined based on playing the arms according to a static optimal distribution, while our dynamic benchmark $OPT(T)$ allows the arm plays following a different distribution over time. For the stochastic BwK problem, these two are equivalent. In fact, our dynamic benchmark is more relevant in the application contexts such as AdWords and pricing problem. For example, in practice, one hopes to have a performance guarantee against a benchmark where the airlines/hotels/ski-resorts can price differently for different dates, and the Ads platforms can perform a different allocation rule based on the remaining budgets of the Ads bidders (rather than to compete against a fixed price). While the static benchmark is reasonable for MAB problems motivated from a learning theory context -- one may want to compete against the single (static) best ML expert among all the experts, it is not so much for the BwK problem.
> - More importantly, even against the weaker static benchmark, the existing adversarial BwK only gives a $O(log T)$ competitiveness, i.e. a guarantee of $O(OPT_{static}/\log T)$ achieved reward. Putting aside the persuasiveness of such guarantee in a practical context, the result is in contrast with the MAB problem where EXP3 algorithm achieves $O(\sqrt{T})$ regret, i.e. a guarantee of $OPT_{static}-O(\sqrt{T})$ achieved reward. Our paper attributes this contrast to presence of the constraint. And with the help of the new non-stationarity measures $W_1$ and $W_2,$ we align the regret bound for the BwK problem with that of the MAB problem (Besbes et al., 2014). As a side product, we show such measure is also critical for the OCO problem with constraint.
> - To follow up, our regret bound serves as a problem-dependent regret bound for the adversarial BwK problem. It relates the algorithm performance with the intensity of the adversity (measured by the non-stationarity measures). Such guarantee is more meaningful in practical contexts such as the pricing problem, where the marketing environment will not really play adversarially against the hotels and airlines. In comparison, the adversarial BwK aims to capture the “most” adversarial case of the underlying environment, the regret bound of which can be too conservative.
>
> We conduct more numerical experiments and update our supplementary material. We really appreciate if you have time to read them. The numerical experiments show that the stochastic BwK algorithm can fail drastically even when the underlying distribution changes only once over time. Our algorithm, though not much novelty in its design idea, performs consistently better than both the stochastic BwK algorithm and the adversarial BwK algorithm.
>
> We look forward to your further feedback and will provide our response/clarification timely in the following week.

---

### Meta-Review · Area_Chair_BENS · 2022-09-01

**Recommendation:** Accept
**Confidence:** Less certain

**Metareview:**

The paper provides a nontrivial extension of dynamic regret minimization to knapsack bandit problems in the mixed (obliviously adversarially chosen distributions), and identifies new measures of complexity for constrained decision making problems. Its algorithmic novelty is rather limited, though, as it borrows the 'usual' ideas of windowed averaging with statistical upper/lower bounds for enabling 'forgetting' which is crucially required in nonstationary environments.

The reviewers for the paper are largely appreciative of the paper's contributions in developing and arguing for new measures of nonstationarity, in terms of dependence on both the constraint and the reward side. Their concerns have been responded to in detail by the author(s), explaining both technical and motivation-related aspects on one hand, and connections to existing related work on the other. With the belief that this line of work may spur other creative directions in constrained decision-making / bandit problems, I am happy to recommend acceptance. I urge the author(s) to use the extra page of space to incorporate their responses to many of the reviewers' queries.

**Award:**

No

---

### Decision · Program_Chairs · 2022-09-14

Accept